# Three-dimensional growth of breast cancer cells potentiates the anti-tumor effects of unacylated ghrelin and AZP-531

CheukMan C Au[1,2,3], John B Furness[4], Kara Britt[5,6], Sofya Oshchepkova[1], Heta Ladumor[1,7], Kai Ying Soo[2], Brid Callaghan[4], Celine Gerard[2], Giorgio Inghirami[8], Vivek Mittal[9], Yufeng Wang[10], Xin Yun Huang[10], Jason A Spector[11], Eleni Andreopoulou[1], Paul Zumbo[10,12], Doron Betel[1,13], Lukas Dow[1], Kristy A Brown[1,2,3]*

[1]Department of Medicine, Weill Cornell Medicine, New York, United States; [2]Centre for Cancer Research, Hudson Institute for Medical Research, Clayton, Australia; [3]Department of Molecular and Translational Sciences, Monash University, Clayton, Australia; [4]Department of Anatomy and Neuroscience, University of Melbourne, Parkville, Australia; [5]Peter MacCallum Cancer Centre, Melbourne, Australia; [6]Sir Peter MacCallum Department of Oncology, University of Melbourne, Melbourne, Australia; [7]Weill Cornell Medicine - Qatar, Doha, Qatar; [8]Department of Pathology, Weill Cornell Medical College, New York, United States; [9]Department of Cardiothoracic Surgery, Department of Cell and Developmental Biology, Neuberger Berman Lung Cancer Center, Weill Cornell Medicine, New York, United States; [10]Department of Physiology and Biophysics, Weill Cornell Medical College of Cornell University, New York, United States; [11]Department of Surgery, Weill Cornell Medicine, New York, United States; [12]Applied Bioinformatics Core, Weill Cornell Medical College, New York, United States; [13]Institute for Computational Biomedicine, Weill Cornell Medical College, New York, United States

*For correspondence:
kab2060@med.cornell.edu

**Abstract** Breast cancer is the most common type of cancer in women and notwithstanding important therapeutic advances, remains the second leading cause of cancer-related death. Despite extensive research relating to the hormone ghrelin, responsible for the stimulation of growth hormone release and appetite, little is known of the effects of its unacylated form, especially in cancer. The present study aimed to characterize effects of unacylated ghrelin on breast cancer cells, define its mechanism of action, and explore the therapeutic potential of unacylated ghrelin or analog AZP-531. We report potent anti-tumor effects of unacylated ghrelin, dependent on cells being cultured in 3D in a biologically-relevant extracellular matrix. The mechanism of unacylated ghrelin-mediated growth inhibition involves activation of Gαi and suppression of MAPK signaling. AZP-531 also suppresses the growth of breast cancer cells *in vitro* and in xenografts, and may be a novel approach for the safe and effective treatment of breast cancer.

## Introduction

Breast cancer is the most commonly diagnosed cancer in women and second only to lung cancer in terms of mortality (*Noone et al., 2018*). The majority of breast cancers occur after menopause and are hormone receptor positive, and in these women, first-line therapy usually involves endocrine therapy, for example aromatase inhibitors or tamoxifen (*Reinert and Barrios, 2015*). Despite the efficacy of endocrine therapy, a number of women experience severe and debilitating side effects

due to the global inhibition of estrogen biosynthesis or action, and some will cease the use of their potentially life-saving treatment (*Lønning and Eikesdal, 2013*). A proportion of women will also be resistant to treatment or develop resistance over time, and some will have tumors that cannot be treated with targeted therapies, that is triple negative breast cancers (TNBCs). Aggressive breast cancers are often associated with activation of RAS/MAPK signaling (*Giltnane and Balko, 2014*; *Santen et al., 2002*), despite only a minority carrying a mutation in these genes (*Tilch et al., 2014*). The prognosis for these patients is poor and hence, there is a need to identify alternative treatments that are safe and effective.

The unacylated form of ghrelin is closely related to the appetite-stimulating hormone ghrelin. However, because it lacks octanoylation, it does not bind to the cognate ghrelin receptor, GHSR1a. Initially, unacylated ghrelin was believed to be produced as a by-product of ghrelin gene expression. However, recent studies have established an important role for this peptide hormone in regulating energy homeostasis, including reducing fat mass, improving insulin sensitivity and decreasing fasting glucose levels (*Benso et al., 2012*; *Zhang et al., 2008*). An unacylated ghrelin analog, AZP-531 (levolitide), is currently in clinical trials for the treatment of Prader-Willi Syndrome and type II diabetes (*Allas et al., 2016*). AZP-531 has an established safety profile and better pharmacokinetic properties than unacylated ghrelin (*Allas et al., 2016*). The receptor for unacylated ghrelin is currently unknown (*Au et al., 2016*) and consequently, there is an important gap in our understanding of the mechanisms mediating its effects.

We have previously demonstrated that unacylated ghrelin has activity in non-cancer tissue, including adipose stromal cells, where it suppresses the expression of the estrogen-biosynthetic enzyme aromatase, and breast adipose tissue macrophages, where it inhibits the production of inflammatory mediators (*Au et al., 2017*; *Docanto et al., 2014*). Here, we show that the unacylated form of ghrelin is a potent suppressor of breast cancer cell growth, independent of effects on the stroma, and provide a novel mechanism of action via activation of Gαi, suppression of cAMP production, and inhibition of MAPK and Akt signaling. Importantly, the potent effects of unacylated ghrelin are dependent on growth of cells in 3D within a relevant extracellular matrix (ECM). Our findings, and that of others, suggest that tumor cell culture context affects response to therapy and that many translational failures may have resulted from the inappropriate model systems used to date (*Tian et al., 2015*). It also suggests that others may have overlooked many effective therapies due to a lack of response in 2D cultures. We therefore believe that unacylated ghrelin is a prototypic 3D-specific breast cancer cell therapeutic and that characterizing its mechanism of action in a biologically relevant ECM will lead to a better understanding of how the tumor microenvironment affects response to therapy. We also demonstrate consistent effects in patient-derived breast cancer cells and breast cancer xenografts in preclinical models, where both unacylated ghrelin and AZP-531 are effective at causing growth inhibition.

## Results

### Unacylated ghrelin inhibits the 3D growth of breast cancer cells

Previous studies examining the effect of ghrelin and unacylated ghrelin on the growth of breast cancer cells showed little activity at doses below 1 µM (*Cassoni et al., 2001*). Considering the increasingly acknowledged role of the ECM in dictating the biology of tumors *in vivo*, including aggressiveness and response to treatment, we sought to examine the effect of ghrelin and more importantly, unacylated ghrelin, in 3D cultures of breast cancer cells. Although both are hypothesized to bind alternate ghrelin receptors, beneficial effects on energy homeostasis have only been reported for unacylated ghrelin – likely due to ghrelin stimulating insulin resistance via its cognate receptor. The dependence on 3D culture was examined for unacylated ghrelin, where treatment of MCF7 and MDA-MB-468 cells with 100 pM resulted in suppression of cell growth in matrigel and collagen, but not in 2D (*Figure 1—figure supplement 1a and b*). At this dose, no effects on MDA-MB-231 cell growth were observed in either 2D or 3D cultures (*Figure 1—figure supplement 1c*). Inhibition of MCF7 and MDA-MB-468 cell growth was observed at $10^{-18}$-$10^{-10}$ M, with maximal effects observed at picomolar doses (*Figure 1—figure supplement 1d and e*). The degree of inhibition is directly related to the degree of growth stimulation induced by serum or estradiol. Again, no significant effect was observed in MDA-MB-231 cells (*Figure 1—figure supplement 1f*). The effects

of unacylated ghrelin were mimicked by ghrelin, where 100 pM of either ghrelin and unacylated ghrelin were found to suppress the growth of MCF7 breast cancer cells when grown in 3D in matrigel (*Figure 1—figure supplement 1g*). To gain insights into the breadth of effects of unacylated and potential predictors of response, the effect of 100 pM unacylated ghrelin was tested in a panel of breast cancer cell lines grown in 3D, including ER+/PR+/HER2- (MCF7, T47D), ER+/PR+/HER2+ (ZR-75), HER2+ (SKBR3), TNBC (MDA-MB-468, DU4475, MDA-MB-157, Hs578T, MDA-MB-231), and tamoxifen-resistant (LCC2) cells (*Figure 1a–c*). In the presence of serum, unacylated ghrelin significantly inhibited the growth of all breast cancer cells examined except for three of the TNBC cell lines (*Figure 1c*; DU4475, Hs578T and MDA-MB-231). Unacylated ghrelin also suppressed the estradiol-stimulated growth of ER+ breast cancer cells (*Figure 1b*). Responsive cells included those having

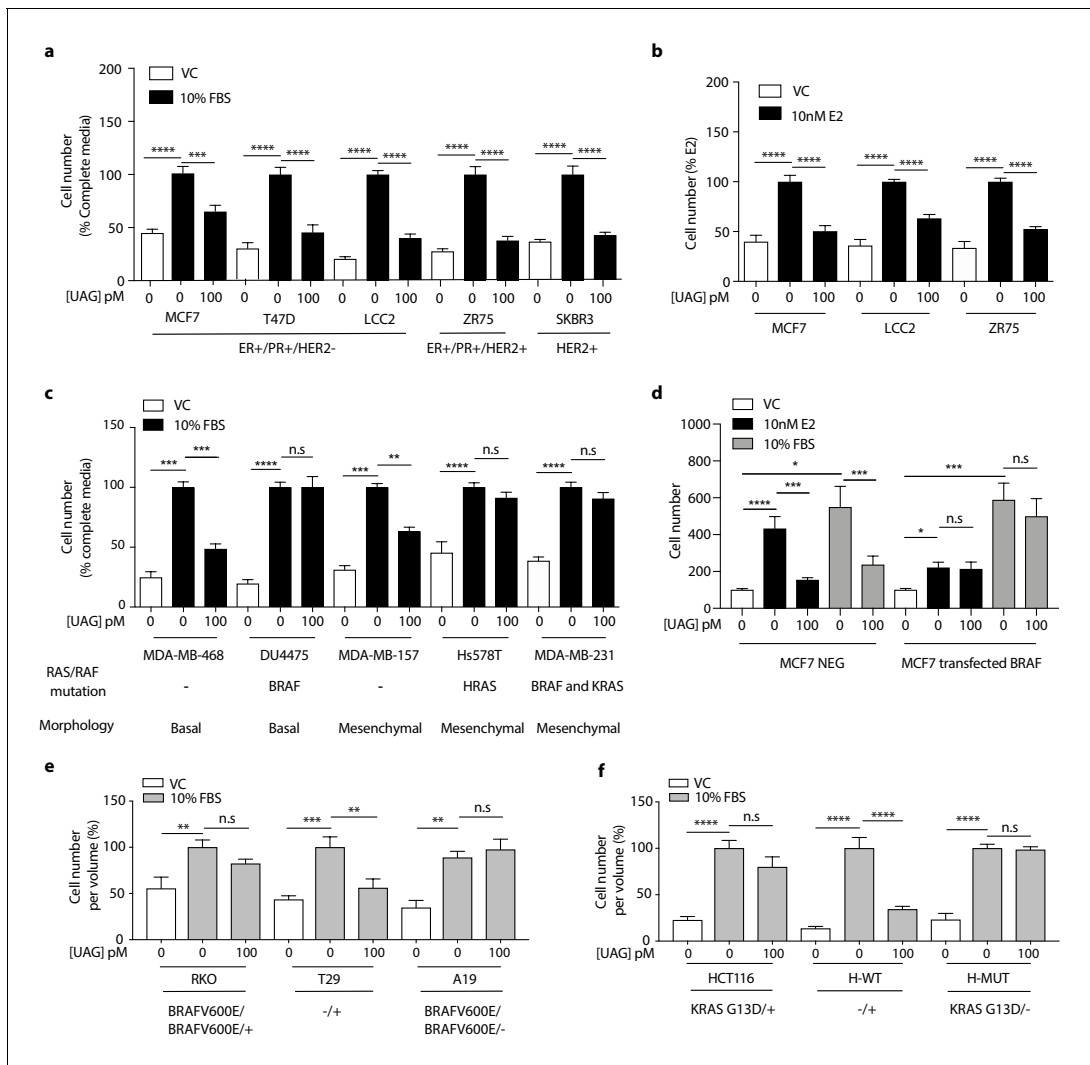

**Figure 1.** Unacylated ghrelin inhibits the 3D growth of breast cancer cells. (**a, c**) Unacylated ghrelin (UAG; 100 pM) inhibits the growth of a panel of breast cancer cell lines under serum-stimulated conditions (six replicates/group) or (**b**) ER+ breast cancer cell lines in the presence of estradiol (10 nM; six replicates/group). (**c**) UAG (100 pM) suppresses cell growth of basal-like and mesenchymal-like TNBC breast cancer cell lines that are WT for *BRAF* and *KRAS* (6–9 replicates/group). Effects of UAG are abrogated in (**d**) BRAF-transfected MCF7 cells, and (**e**) *BRAF-* and (**f**) *KRAS*-mutated colon cancer cells (6–12 replicates/group). Loss of mutated alleles of *BRAF* or *KRAS* sensitizes cells to the effect of UAG. Data represent mean ± SEM. Experiments were repeated at least twice. UAG: unacylated ghrelin; VC: vehicle control; FBS: fetal bovine serum; E2: estradiol.

The online version of this article includes the following source data and figure supplement(s) for figure 1:

**Source data 1.** Unacylated ghrelin inhibits the 3D growth of breast cancer cells.

**Figure supplement 1.** Unacylated ghrelin inhibits the growth of breast cancer cells in 3D.

**Figure supplement 1—source data 1.** Unacylated ghrelin inhibits the growth of breast cancer cells in 3D.

mutations in *PTEN* and *PIK3CA*, while resistant cell lines carried mutations in *BRAF*, *HRAS* and/or *KRAS* (*Figure 1c*; *Table 1*). To test whether these mutations confer resistance to treatment, the effects of unacylated ghrelin were examined in BRAFV600E-transfected MCF7 cells or colon cancer cells (RKO, HCT116) that carry mutations in *BRAF* or *KRAS*, respectively. Transient transfection of MCF7 cells with mutant *BRAF* led to resistance to unacylated ghrelin treatment in estradiol- and serum-stimulated cells (*Figure 1d*). Unacylated ghrelin had no effect on the growth of RKO and HCT116 cells (*Figure 1e and f*). However, loss of the mutant alleles for *BRAF* or *KRAS* led to sensitization of RKO and HCT116 cells, respectively. Loss of the wild-type allele had no significant effect. Binding of Cy3-labeled unacylated ghrelin to responsive and non-responsive cells (*Figure 1—figure supplement 1h*) suggests that resistance is not due to lack of binding or receptor expression.

## Unacylated ghrelin suppresses breast cancer cell growth via Gαi-dependent inhibition of cAMP formation

Unacylated ghrelin does not bind to the cognate ghrelin receptor GHSR1a and the receptor for unacylated ghrelin, believed to be an alternate ghrelin receptor, is currently unknown, but hypothesized to be a GPCR. Effects of unacylated ghrelin on second messenger systems were assessed by measuring the formation of cAMP and the release of intracellular calcium in MCF7 cells (*Figure 2a* and *Figure 2—figure supplement 1a*). Unacylated ghrelin significantly suppressed the formation of cAMP, but had no effect on intracellular calcium release. Effects on cAMP suggest a Gαi-dependent mechanism. Activation of Gαi by unacylated ghrelin was observed after 2 hr treatment, as measured by pull-down of GTP-bound Gαi (*Figure 2b*). To test dependence of growth inhibitory effects of unacylated ghrelin on Gαi, 3D growth assays were performed in cells where the Gαi-encoding gene, guanine nucleotide-binding protein, alpha subunit (*GNAI*), was knocked out (CRISPR) or in the presence of Gαi inhibitor pertussis toxin. There are three Gαi encoding genes in mammalian cells (*GNAI1*, *GNAI2*, and *GNAI3*). Unacylated ghrelin had no effect on MCF7 cells lacking Gαi subunit 2

**Table 1.** Characteristics of breast cancer cell lines and patient-derived breast cancer cells, and responsiveness to unacylated ghrelin.

| Breast cancer cell line/Patient sample | Known mutations | Intrinsic subtype | Receptor status | Responsive to Unacylated Ghrelin |
|---|---|---|---|---|
| **Cell line** | | | | |
| MCF7 | *CDKN2A, PIK3CA* | Luminal A | ER+/PR+/HER2- | Yes |
| LCC2 | N/A | Luminal A | ER+/PR+/HER2- | Yes |
| T47D | *PIK3CA, TP53* | Luminal A | ER+/PR+/HER2- | Yes |
| ZR75 | *PTEN* | Luminal B | ER+/PR+/HER2+ | Yes |
| SKBR3 | *TP53* | HER2+ | HER2+ | Yes |
| MDA-MB-468 | *PTEN, RB1, SMAD4, TP53* | Basal-like | TNBC | Yes |
| MDA-MB-157 | *NF1, TP53* | Mesenchymal-like | TNBC | Yes |
| MDA-MB-231 | *BRAF, KRAS, TP53, CDKN2A, NF2* | Mesenchymal-like | TNBC | No |
| HS578T | *HRAS, TP53* | Mesenchymal-like | TNBC | No |
| DU4475 | *BRAF, APC, MAP2K4, RB1* | Basal-like | TNBC | No |
| **Patient samples** | | | | |
| ER+ Case 1 | N/A | Luminal A | ER+ | Yes |
| ER+ Case 2 | N/A | Luminal A | ER+ | Yes |
| 2147-TG5 | N/A | Basal-like | TNBC | Yes |
| 4013-TG3 | N/A | Basal-like | TNBC | Yes |
| 3887-TG7 | N/A | Mesenchymal-like | TNBC | No |
| 3204-TG6 | N/A | Mesenchymal-like | TNBC | No |

Abbreviations: ER, estrogen receptor; PR, progesterone receptor; HER2, human epidermal growth factor receptor 2; TNBC, triple negative breast cancer; N/A: not available.

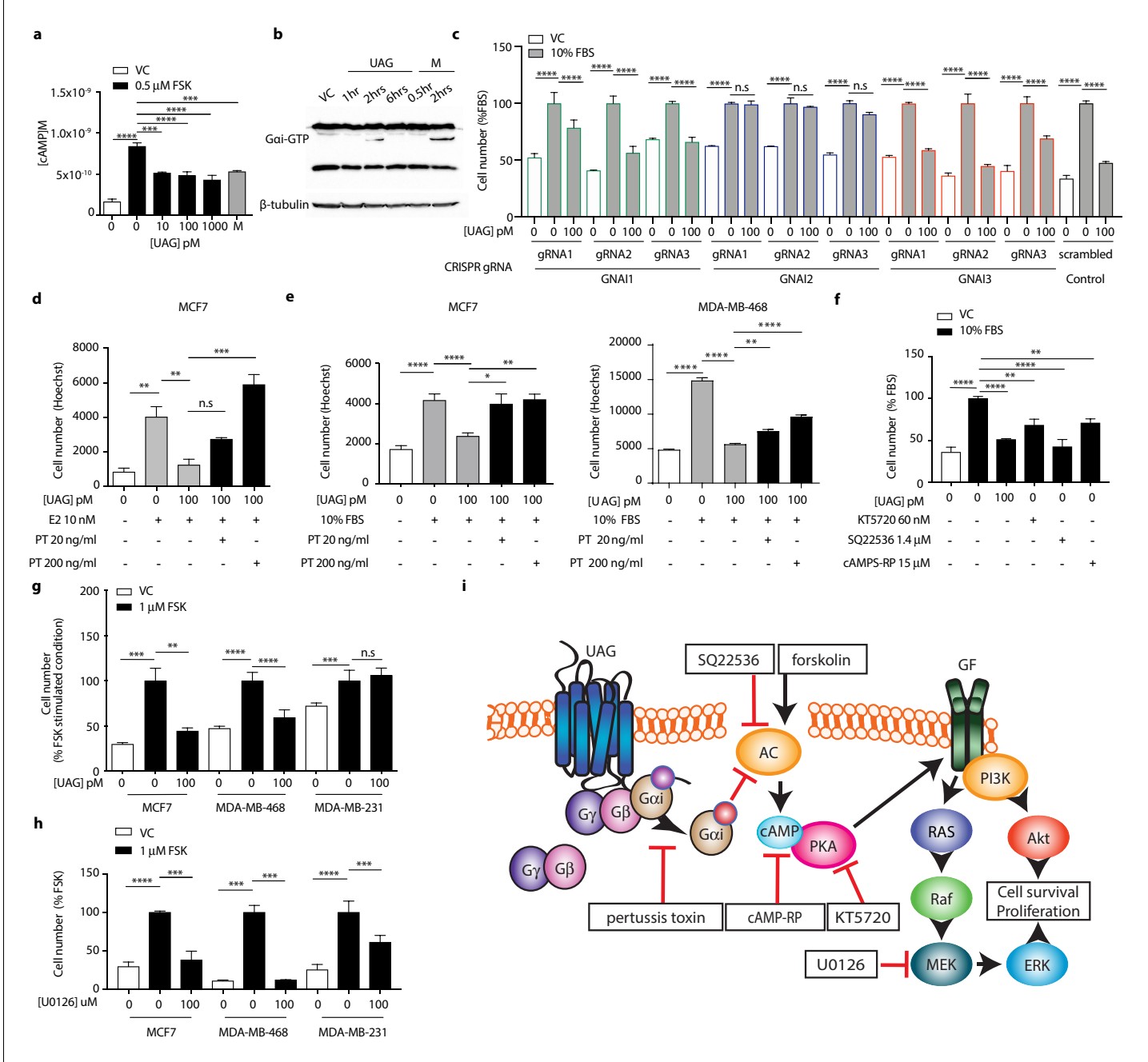

**Figure 2.** Unacylated ghrelin suppresses breast cancer cell growth via Gαi-dependent inhibition of cAMP formation. (**a**) Unacylated ghrelin (UAG; 10–1000 pM) inhibits the forskolin-stimulated production of cAMP in MCF7 cells (3–4 replicates/group). (**b**) UAG (100 pM) stimulates activation of Gαi (three replicates/group). (**c**) UAG (100 pM) suppresses the growth of CRISPR *GNAI1* and *GNAI3* KO cells, but not *GNAI2* KO MCF7 cells, suggesting Gαi2-coupled GPCR-mediated effects. Suppression of (**d**) estradiol- or (**e**) serum-stimulated breast cancer cell growth with UAG (100 pM) is prevented in the presence of Gαi inhibitor, pertussis toxin (20 ng/ml, 200 ng/ml; three replicates/group). (**f**) PKA inhibitor (KT5720), adenylyl cyclase inhibitor (SQ22536) or cAMP antagonist (cAMPS-RP) suppress the serum-stimulated growth of MCF7 cells (three replicates/group). (**g**) UAG (100 pM) inhibits the forskolin-stimulated growth of MCF7 and MDA-MB-468 cells, but not MDA-MB-231 (three replicates/group). (**h**) U0126 (MEK inhibitor) inhibits the forskolin-stimulated growth of MCF7, MDA-MB-468 and MDA-MB-231 cells (three replicates/group). (**i**) A model summarizing the putative mechanism of action of UAG in breast cancer cells and compounds used to dissect mechanism of action. Data represent mean ± SEM. Experiments were repeated at least twice. UAG: unacylated ghrelin; VC: vehicle control; FBS: fetal bovine serum; E2: estradiol; PT: pertussis toxin; M: melatonin; FSK: forskolin. The online version of this article includes the following source data and figure supplement(s) for figure 2:

**Source data 1.** Unacylated ghrelin suppresses breast cancer cell growth via Gαi-dependent inhibition of cAMP formation.

**Source data 2.** Unacylated ghrelin suppresses breast cancer cell growth via Gαi-dependent inhibition of cAMP formation.

*Figure 2 continued on next page*

*Figure 2 continued*

**Figure supplement 1.** Unacylated ghrelin suppresses breast cancer cell growth via Gαi-dependent mechanisms.

**Figure supplement 1—source data 1.** Unacylated ghrelin suppresses breast cancer cell growth via Gαi-dependent mechanisms.

(*GNAI2*), whereas the growth of cells was significantly suppressed where Gαi subunit 1 (*GNAI1*) and Gαi subunit 3 (*GNAI3*) were targeted (*Figure 2c*). *GNAI2* was also found to be required for unacylated ghrelin activity in MDA-MB-468 cells (*Figure 2—figure supplement 1b*). Effects of unacylated ghrelin were also attenuated in MCF7, MDA-MB-468 and ZR-75 cells treated with pertussis toxin, in the presence of estradiol or serum (*Figure 2d and e* and *Figure 2—figure supplement 1c*). To determine whether inhibition of cAMP is sufficient to suppress the serum-stimulated growth of breast cancer cells, MCF7 and MDA-MB-468 cells were treated with adenylyl cyclase and PKA inhibitors, SQ22536 and KT5720, and cAMP antagonist, cAMPS-RP (*Figure 2f*; *Figure 2—figure supplement 1d*). Inhibition of cAMP formation and PKA led to a significant reduction in cell number. Effects of cAMP on cell growth were then examined in cells treated with adenylyl cyclase stimulator, forskolin (*Figure 2g* and *Figure 2—figure supplement 1e*). Forskolin stimulated the growth of MCF7, ZR75, MDA-MB-468 and MDA-MB-231 cells, and similar to effects of unacylated ghrelin in serum- or estradiol-stimulated conditions, unacylated ghrelin suppressed the forskolin-mediated induction of MCF7, ZR75 and MDA-MB-468 cell growth, but not MDA-MB-231. As resistance to unacylated ghrelin was observed in cells that carry mutations in RAS and RAF, the link between cAMP and MAPK signaling was next examined in MCF7, MDA-MB-468 and MDA-MB-231 cells in 3D (*Figure 2h*). Inhibition of MEK activity using U0126 led to a significant reduction in the forskolin-stimulated growth of all cell lines.

## Unacylated ghrelin inhibits MAPK and Akt signaling

Since unacylated ghrelin suppresses cAMP, which we found to stimulate cell growth via MAPK-dependent mechanisms, we next sought to determine whether unacylated ghrelin affected MAPK signaling in 3D cultures, both acutely and chronically. Effects of unacylated ghrelin on ERK activity were examined in real-time using time-lapse confocal microscopy of the FRET-based extracellular signal-regulated kinase activity reporter (EKAR) in MCF7 cells (*Figure 3a*). Serum stimulated EKAR activity, whereas unacylated ghrelin and the MEK inhibitor, U0126, inhibited this effect. Effects of unacylated ghrelin on MAPK signaling were then examined. Unacylated ghrelin caused a decrease in the phosphorylation of ERK and downstream target, p90RSK (*Figure 3b*). The effect on p90RSK was sustained for 24 hr in MCF7 and MDA-MB-468 cells, but this suppression was not observed in MDA-MB-231 cells, at any time point (not shown). The levels of MAPK target cMyc, induced by serum, were also suppressed in cells treated with unacylated ghrelin (*Figure 3c*). Effects on Akt phosphorylation and activity were then examined by immunoblotting and by quantifying levels of FoxO3 nuclear localization in 3D in real time (*Figure 3d and e*). Unacylated ghrelin inhibited the serum-stimulated phosphorylation of Akt (Ser473) and downstream target p70S6K (Thr389). Inhibition of Akt phosphorylation was noticeable at 5 and 15 min, and sustained for 24 hr. Effects on Akt activity, measured by examining the degree of serum-stimulated FoxO3 nuclear exclusion, were observed in real time, with unacylated ghrelin and PI3K inhibitor (LY294002) causing a significant reduction in FoxO3 nuclear exclusion after 1 hr treatment. No effect of unacylated ghrelin on Akt phosphorylation or activity was observed in MDA-MB-231 cells (not shown). These data suggest that unacylated ghrelin inhibits cell proliferation via effects on MAPK and Akt signaling.

## Unacylated ghrelin causes cell cycle arrest and apoptosis

Additional studies were then undertaken to determine whether effects of unacylated ghrelin on cell number were due to stimulation of cell cycle arrest or apoptosis, or both. The effects of unacylated ghrelin on cell proliferation were examined by assessing EdU incorporation in breast cancer cell lines grown in 3D. Estradiol and serum caused a significant increase in EdU incorporation, and unacylated ghrelin suppressed DNA synthesis at 10, 100 and 1000 pM in MCF7, ZR75 and tamoxifen-resistant LCC2 cells (*Figure 4a–c*, *Figure 4—figure supplement 1a*). The effect of unacylated ghrelin on phases of the cell cycle was assessed in live cells in 3D using the Premo FUCCI Cell Cycle Sensor, which consists of RFP-tagged cdt1 and GFP-tagged geminin (*Figure 4d–e*, and *Figure 4—figure*

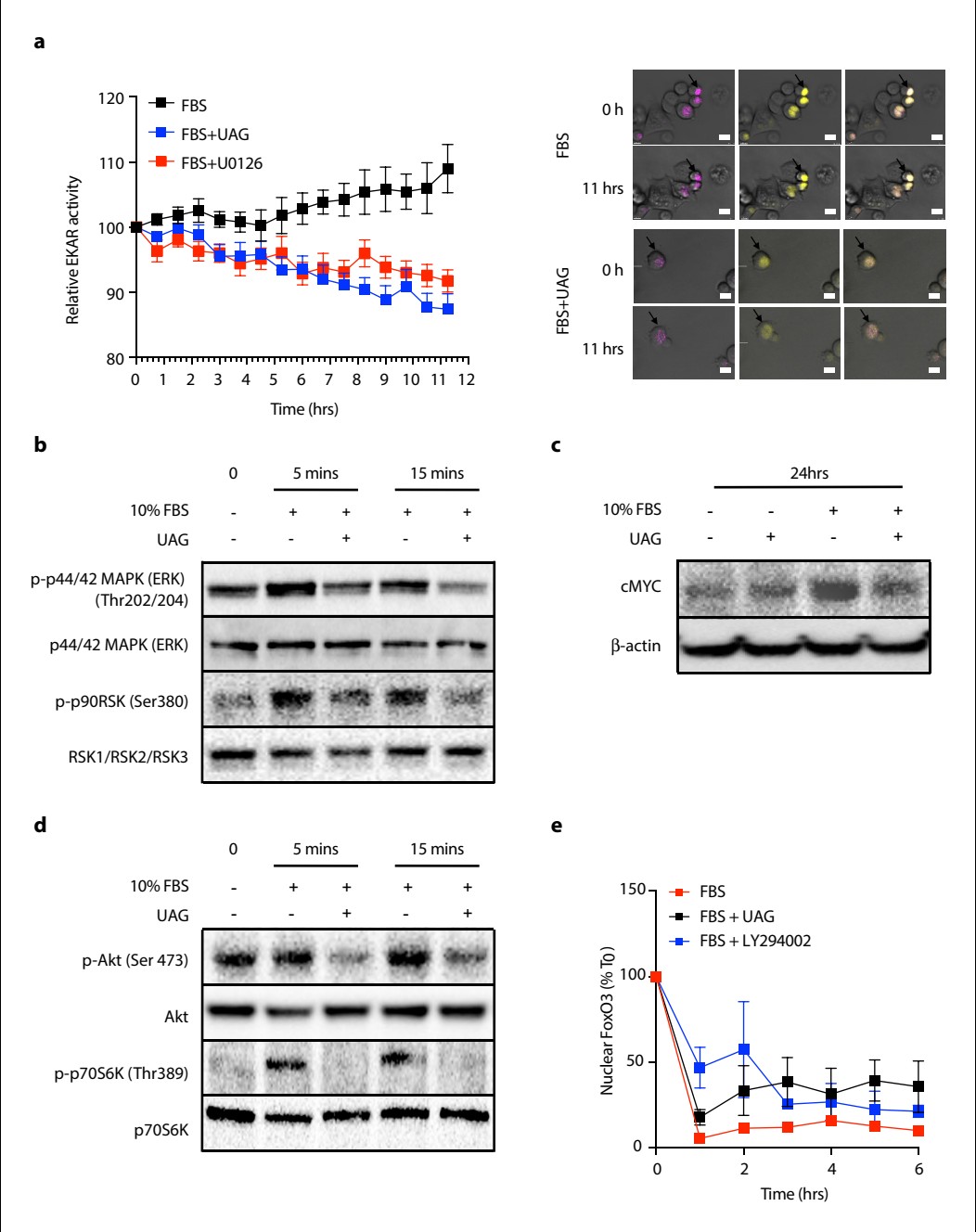

**Figure 3.** Unacylated ghrelin (UAG) inhibits MAPK and Akt signaling. (**a**) Unacylated ghrelin (UAG; 100 pM) inhibits ERK activity (EYFP FRET) in EKAR-transfected MCF7 cells (10 replicates/group). Data were normalized to vector ECFP signal. Scale bar represent 50 μm. Western blotting demonstrates that UAG causes a decrease in the (**b**) phosphorylation of ERK1/2 and downstream MAPK target p90RSK and (**c**) expression of cMYC in MCF7 cells. UAG also causes a decrease in (**d**) the phosphorylation of Akt and its downstream target, p70S6K, as well as (**e**) FoxO3a nuclear localization FoxO3a-RFP-transfected cells, an effect that is attenuated in cells treated with PI3K inhibitor LY294002 (five replicates/group). Data represent mean ± SEM. Experiment were repeated at least twice. UAG: unacylated ghrelin; FBS: fetal bovine serum.

The online version of this article includes the following source data for figure 3:

**Source data 1.** Unacylated ghrelin suppresses EKAR and FoxO3 nuclear localization.

**Source data 2.** Unacylated ghrelin inhibits MAPK and Akt signaling.

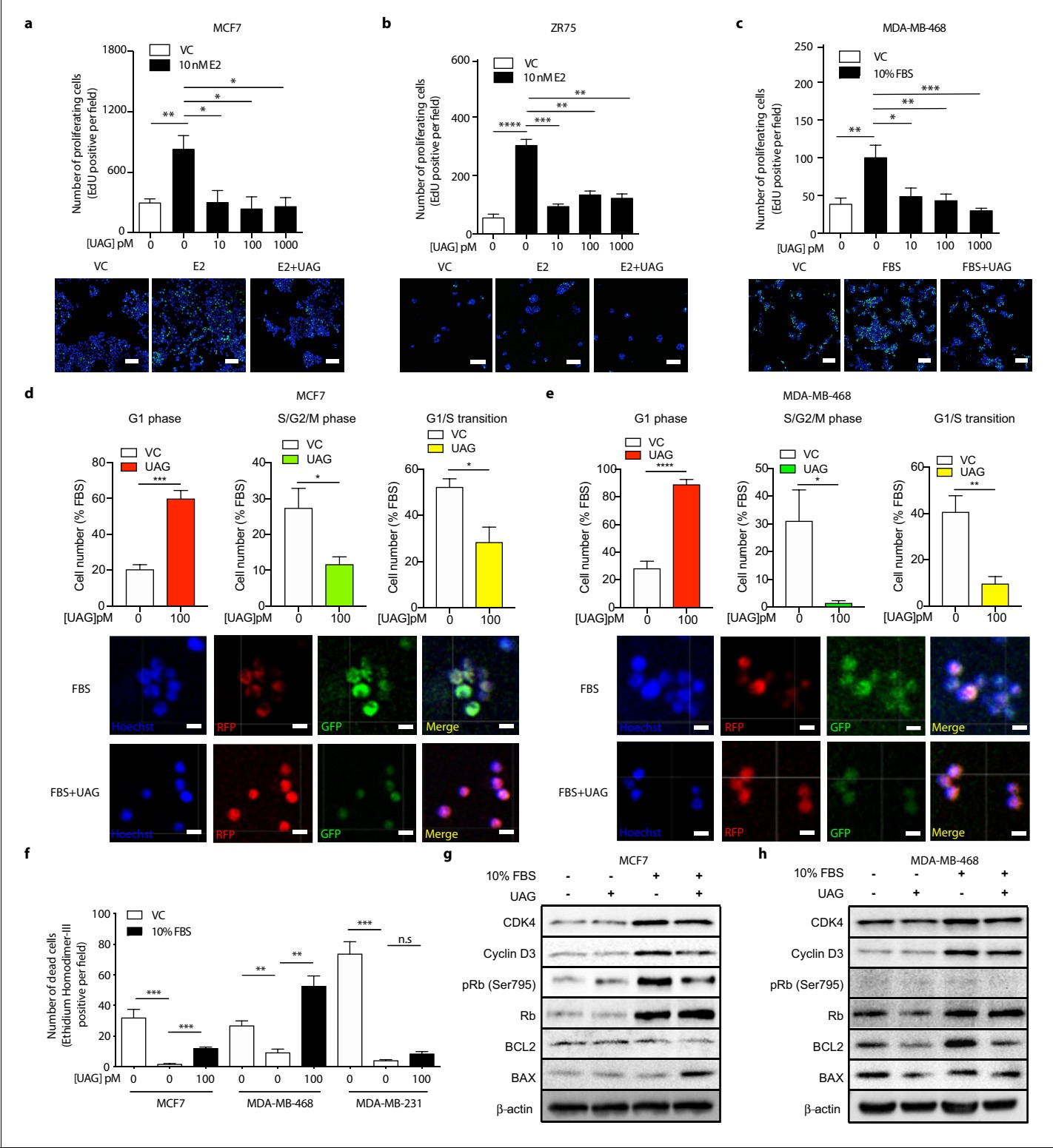

**Figure 4.** Unacylated ghrelin causes cell cycle arrest and apoptosis. Unacylated ghrelin (UAG) significantly inhibits the proliferation of (**a**) MCF7, (**b**) ZR75 and (**c**) MDA-MB-468 in the presence of estradiol (10 nM) or serum (3–6 replicates/group). Representative images showing EdU incorporation (green). Hoechst nuclear stain; blue. Scale bar represent 100 µm. Effects are mediated via induction of G1-phase cell cycle arrest (RFP+) and a reduction in the number of cells in S/G2/M-phase (GFP+) and G1/S transition (YFP+) in (**d**) MCF7 and (**e**) MDA-MB-468 cells (four replicates/group). (Hoechst nuclear stain; blue). Scale bar represents 100 µm. (**f**) UAG stimulates cell death in MCF7 and MDA-MB-468 cells, but not MDA-MB-231 cells (three

*Figure 4 continued on next page*

Figure 4 continued

replicates/group). Western blot results demonstrating that UAG inhibits CDK4/cyclin D3, pRB (Ser 795) and BCL2, and stimulates BAX in (**g**) MCF7 and (**h**) MDA-MB-468 cells. Data represent mean ± SEM. Experiments were repeated at least twice. UAG: unacylated ghrelin; VC: vehicle control; FBS: fetal bovine serum; E2: estradiol.

The online version of this article includes the following source data and figure supplement(s) for figure 4:

**Source data 1.** Unacylated ghrelin causes cell cycle arrest and apoptosis.
**Source data 2.** Unacylated ghrelin causes cell cycle arrest and apoptosis.
**Figure supplement 1.** Unacylated ghrelin causes cell cycle arrest and apoptosis.
**Figure supplement 1—source data 1.** Unacylated ghrelin causes cell cycle arrest and apoptosis.

*supplement 1b*). In G1, geminin is degraded, leaving only the RFP signal. In S, G2 and M phases, cdt1 is degraded and hence, only the GFP signal is detected. During the G1/S transition, cells express both proteins with overlapping fluorescence appearing as yellow. Treatment of MCF7 and MDA-MB-468 cells with unacylated ghrelin was associated with an increase in cells arrested at the G1 phase of the cell cycle, with no significant effect in MDA-MB-231 cells. Unacylated ghrelin also caused apoptosis in serum-stimulated MCF7 and MDA-MB-468 cells, measured in 3D using membrane-impermeant nucleic acid dye ethidium homodimer III and using FACS analysis of Annexin V-stained cells (*Figure 4f*; *Figure 4—figure supplement 1*). Consistently, unacylated ghrelin suppressed the serum-stimulated expression of CDK4 and cyclin D3, important for cell cycle G1 phase progression, and decreased phosphorylation of Rb at Ser795, known to prevent the degradation of this cell cycle arrest protein. Treatment with unacylated ghrelin was also associated with a decreased expression of anti-apoptotic protein BCL2, while stimulating the expression of pro-apoptotic factor BAX (*Figure 4g–h*). This effect was not observed in MDA-MB-231 cells (not shown). These data therefore suggest that unacylated ghrelin decreases cell number by stimulating cells to arrest in G1 and stimulating apoptosis.

## Unacylated ghrelin inhibits tumor growth in xenograft models and patient-derived tumor cells

The effect of unacylated ghrelin on tumor growth *in vivo* was examined in orthotopic xenograft and allograft mouse models. In MCF7 and ZR75 xenografts, daily s.c. injection of 50 µg/kg and 100 µg/kg unacylated ghrelin led to a significant reduction in tumor volume (*Figure 5a and b*). Treatment with unacylated ghrelin had no detrimental effect on weight and no abnormalities associated with treatment were observed with histopathology assessment of a subset of mice. Effects were then assessed in a syngeneic model of mammary cancer. In the J110 allograft model, unacylated ghrelin caused a significant reduction in tumor growth at 100 µg/kg and 200 µg/kg (*Figure 5c*). Effects of unacylated ghrelin to suppress the growth of J110 cells was not dependent on the host immune context, as similar results were obtained when cells were xenografted in Balb/c nude immunocompromised mice (*Figure 5—figure supplement 1d*). The degree of apoptosis in tumors at endpoint was quantified by counting the percentage of cells with pyknotic nuclei in tumors. Unacylated ghrelin significantly increased the number of apoptotic cells in MCF7, ZR75 and J110 xenografts/allografts (*Figure 5d–f*). The effects of unacylated ghrelin were also examined in patient-derived breast cancer cells. Unacylated ghrelin at 100pM caused a significant reduction in the serum-induced growth of ER + breast cancer cells, while inhibiting or having no effect in the TNBC patient samples examined (*Figure 5g*, *Figure 5—figure supplement 1e*; *Table 1*). BRAF and KRAS mutation status has not been characterized in these patient samples. Interestingly, Ingenuity Pathway Analysis (causal analysis) of baseline gene expression indicated that MAPK-target genes were differentially expressed between responsive and non-responsive TNBC patient-derived samples, such that non-responsive cells had gene expression consistent with activation of MAPK signaling. This gene signature may allow prediction of responders in a patient population.

## Unacylated ghrelin analog, AZP-531, inhibits breast cancer cell growth *in vitro*, *ex vivo* and *in vivo*

AZP-531 is a cyclic analog of unacylated ghrelin comprised of amino acids $Ser_6-Gln_{12}$ of the unacylated ghrelin peptide. It is a well-tolerated drug that is more stable than unacylated ghrelin. In order

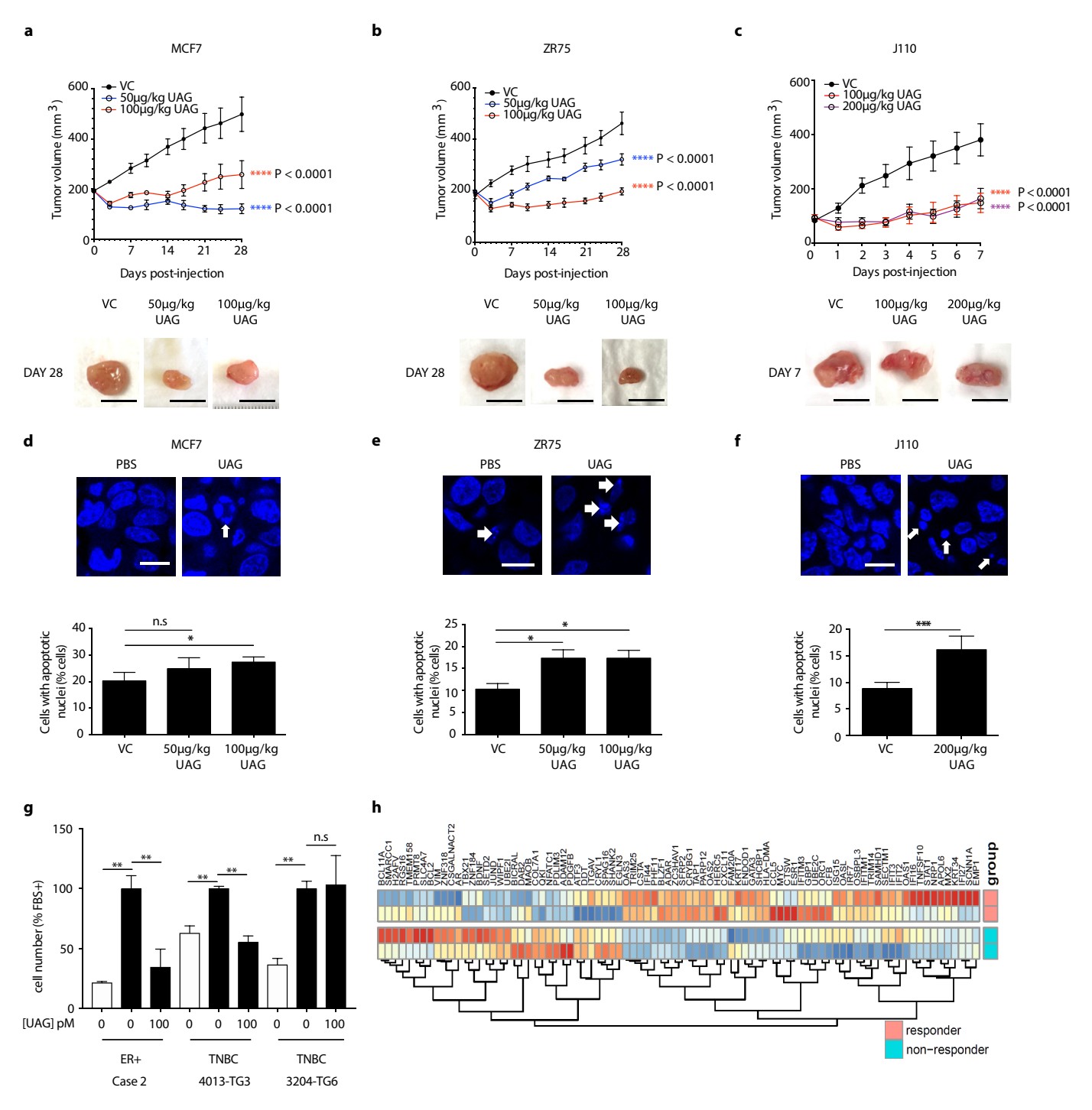

**Figure 5.** Unacylated ghrelin inhibits tumor growth in xenograft models and patient-derived tumor cells. Tumor volume in response to treatment with 50 µg/kg (blue), 100 µg/kg (red) or 200 µg/kg (purple) UAG in mice xenografted with (**a**) MCF7 (six replicates/group), (**b**) ZR75 (five replicates/group), or allografted with (**c**) J110 (five replicates/group) cells. Representative tumor (below) with scale bar representing 10 mm. UAG significantly increases the number of cells with apoptotic nuclei in (**d**) MCF7, (**e**) ZR75 and (**f**) J110 xenografts. (**g**) UAG (100 pM) significantly inhibits the growth of patient-derived ER+ breast cancer cells and 4013-TG3 TNBC cells, but not 3204-TG6 TNBC cells. (**h**) Heatmap representing baseline differential expression of MAPK-target genes in responsive vs. non-responsive patient-derived cells. Data represent mean ± SEM. UAG: unacylated ghrelin; VC: vehicle control; FBS: fetal bovine serum.

The online version of this article includes the following source data and figure supplement(s) for figure 5:

**Source data 1.** Unacylated ghrelin and cyclic analog AZP-531 inhibit tumor growth in xenograft models and patient-derived tumor cells.

*Figure 5 continued on next page*

*Figure 5 continued*

**Figure supplement 1.** Unacylated ghrelin and cyclic analog AZP-531 inhibit tumor growth in xenograft models and patient-derived tumor cells.
**Figure supplement 1—source data 1.** Unacylated ghrelin and cyclic analog AZP-531 inhibit tumor growth in xenograft models andpatient-derived tumor cells.
**Figure supplement 2.** Expression pattern of MAPK-target genes in responder and non-responder TNBC patient-derived breast cancer cases.

to determine whether this clinically available treatment has similar effects to parent peptide unacylated ghrelin, the effect of AZP-531 on breast cancer cell growth was examined in MCF7, MDA-MB-468, and patient-derived TNBC cells, and compared to chemotherapeutic doxorubicin (*Figure 6a–c*). AZP-531 caused the dose-dependent inhibition of the serum-stimulated growth of cells with greater potency than doxorubicin, but unlike doxorubicin, did not reduce cell number beyond that which was stimulated by serum. In order to confirm a similar mechanism of action, the effects of unacylated ghrelin and AZP-531 on cAMP levels within MCF7 cells were compared (*Figure 6d*). Both unacylated ghrelin and AZP-531 caused a dose-dependent inhibition of the forskolin-mediated production of cAMP at 100 and 1000 pM, with no significant differences observed when both

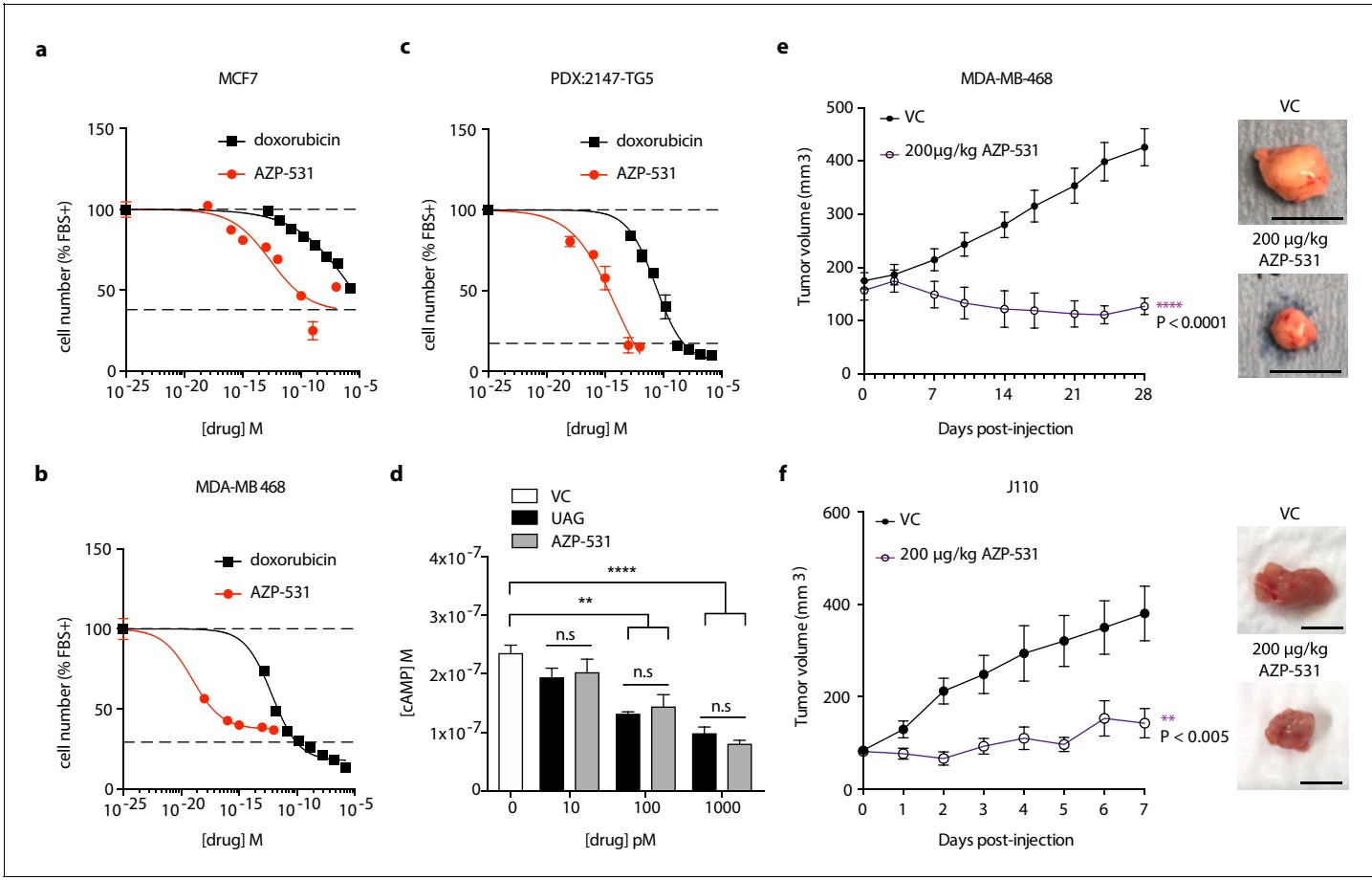

**Figure 6.** Unacylated ghrelin analog, AZP-531, inhibits breast cancer cell growth *in vitro*, *ex vivo* and *in vivo*. AZP-531 causes the dose-dependent inhibition of (**a**) MCF7 and (**b**) MDA-MB-468 and (**c**) patient-derived TNBC breast cancer cell growth in 3D, compared with chemotherapeutic agent doxorubicin (three replicates/group). (**d**) Unacylated ghrelin (UAG; 10–1000 pM) and AZP-531 (AZP; 10–1000 pM) inhibits the forskolin-stimulated production of cAMP in MCF7 cells (3–6 replicates/group). Data represent mean ± SEM. Experiments were repeated at least twice. Tumor volume in response to treatment with 200 µg/kg AZP-531 (purple) in mice xenografted with (**e**) MDA-MB-468 (8–9 replicates/group) or allografted with (**f**) J110 (five replicates/group) cells. Representative images (below) with scale bars representing 10 mm.
The online version of this article includes the following source data for figure 6:

**Source data 1.** Unacylated ghrelin analog, AZP-531, inhibits breast cancer cell growth *in vitro*, *ex vivo* and *in vivo*.

treatments were compared. Importantly, AZP-531 also caused the suppression of cell growth in preclinical models, including TNBC MDA-MB-468 xenografts and ER+ J110 allografts, at 200 µg/kg (*Figure 6e and f*, respectively, and *Figure 5—figure supplement 1*). These data suggest that AZP-531 may be efficacious at suppressing the growth of ER+ tumors and TNBC in women.

## Discussion

Our work provides evidence that unacylated ghrelin is a potent inhibitor of breast cancer cell growth and provides mechanistic insights not previously described for this peptide hormone. Little is known of the relationship between ghrelin, unacylated ghrelin and effects on breast cancer risk and progression. One report recently demonstrated that tumor ghrelin expression is associated with a favorable outcome in invasive breast cancer (*Grönberg et al., 2012*). More specifically, ghrelin immunoreactivity is significantly correlated with low histological grade, estrogen receptor positivity, small tumor size, low proliferation, as well as better recurrence-free and breast cancer-specific survival. The polyclonal antibody used was generated against full-length human ghrelin and hence potentially cross-reacts with unacylated ghrelin. In the present manuscript, both ghrelin and unacylated ghrelin were found to cause potent inhibition of breast cancer cell growth at picomolar doses when cells are grown in a biologically relevant extracellular matrix (ECM). Previous *in vitro* work examining the effects of ghrelin and unacylated ghrelin in breast cancer saw very little effects at submicromolar doses (*Cassoni et al., 2001*). This is likely due to prior studies having been undertaken using traditional culture methods where cells were grown in 2D on plastic and highlights important mechanistic differences when cells are cultured in 3D surrounded by an ECM. The importance of modeling breast and other cancers using 3D systems has been emphasized in a number of recent studies. Importantly, differences in signaling pathway activation are noted and 3D systems are hypothesized to allow for better prediction of *in vivo* effects (*Gangadhara et al., 2016*; *Weigelt et al., 2014*). Our findings, and that of others, suggest that tumor cell culture context affects response to therapy and that many translational failures may have resulted from the inappropriate model systems used to date. It also suggests that others may have overlooked many effective therapies due to a lack of response in 2D cultures.

The cognate ghrelin receptor, GHSR1a, does not bind to unacylated ghrelin and is undetectable in most breast cancer cell lines (*Callaghan and Furness, 2014*; *Cassoni et al., 2001*), suggesting that both ghrelin and unacylated ghrelin are acting at an alternate ghrelin receptor. In order to start dissecting the mechanism of action of unacylated ghrelin, cues were taken from our previous studies in adipose stromal cells where treatment results in inhibition of cAMP (*Docanto et al., 2014*). Results in the present study are consistent with these previous findings, and further demonstrate that unacylated ghrelin treatment is associated with activation of Gαi. The evidence for the importance of cAMP in breast cancer biology has been inconsistent. A number of studies have demonstrated pro-proliferative effects (*Aronica et al., 1994*; *Küng et al., 1983*), while others show that cAMP and activation of PKA inhibit cell growth (*Chen et al., 1998*; *Cho-Chung et al., 1981*; *Fontana et al., 1987*). Effects appear to be cell and context dependent. Directionality also seems more robust for studies performed using clinical samples. For example, cAMP levels have been shown to be 15 times higher in human breast cancer compared to normal breast tissue (*Minton et al., 1974*) and the overall ability to hydrolyze cyclic nucleotides has been shown to be decreased in faster growing and more invasive mammary cancers (*Fajardo et al., 2014*). PKA regulatory subunit expression and catalytic activity have also been shown to be increased in malignant compared to normal breast tissue (*Gordge et al., 1996*), suggesting that cAMP and PKA promote breast cancer growth. Here, we demonstrate that inhibition of cAMP formation or action mimics the effects of unacylated ghrelin to inhibit breast cancer cell growth in 3D. Our findings demonstrating that cells are insensitive to the effects of unacylated ghrelin in the presence of a *BRAF* or *KRAS* mutation also suggests that unacylated ghrelin acts upstream of these signaling proteins. Consistently, the growth-stimulatory effects of cAMP are dependent on MAPK signaling, known to be tightly regulated by BRAF/KRAS signaling.

Effects of unacylated ghrelin to inhibit cell cycle progression and stimulate apoptosis are also consistent with inhibition of MAPK and Akt signaling. Growth factor signaling has previously been shown to cause the accumulation of myc and cyclin D proteins, key drivers of the G1-to-S phase transition, as well as lead to the activation of CDK4 and the hyperphosphorylation of Rb (*Hipfner and Cohen, 2004*; *O'Leary et al., 2016*). Because cell number is not suppressed beyond what is stimulated by

mitogens, effects are likely dependent on signaling downstream of the growth stimulus. The activity of unacylated ghrelin and unacylated ghrelin analogs to suppress MAPK and Akt signaling, myc, cyclin D3 and CDK4 expression, could therefore be leveraged in a therapeutic setting.

The unacylated ghrelin analog, AZP-531, was shown to be well tolerated in a phase I clinical trial performed in overweight and obese subjects with type II diabetes, and significant improvements in glucose variables were observed (*Allas et al., 2016*). A phase II study was also undertaken in individuals with Prader-Willi Syndrome, where daily subcutaneous injections of AZP-531 for 14 days caused a significant improvement in scores on the hyperphagia questionnaire, and a reduction in waist circumference, fat mass and post-prandial glucose levels (*Allas et al., 2018*). We have demonstrated efficacy of AZP-531 in reducing the growth of breast cancer cell lines and patient-derived cancer cells, consistent with effects of unacylated ghrelin. Based on the data obtained using a panel of breast cancer cell lines and effects seen in patient-derived tumor cells, we also identify the subset of breast cancers that will be resistant to treatment, that is TNBCs with *KRAS* or *BRAF* mutations, or TNBCs with high MAPK activity. KRAS has been shown to maintain mesenchymal features of TNBCs (*Kim et al., 2015*). As less than 1% of breast tumors carry these mutations (*Tilch et al., 2014*), it is likely that unacylated ghrelin and AZP-531 will be effective in the majority of breast cancers. Since unacylated ghrelin has also been shown to prevent chemotherapy-induced muscle cell death (*Nonaka et al., 2017*), there is potential to combine unacylated ghrelin or AZP-531 with chemotherapy, while also reducing cardiotoxicity.

# Materials and methods

## Experimental model and subject details

### Human breast cancer tissues and preclinical patient derived xenograft models

Patient-derived breast tumors (PDX [*Zhang et al., 2013*]) were obtained from Dr. Giorgio Inghirami and estrogen receptor positive (ER+) breast tumors were obtained from Dr. Eleni Andreopoulou at Weill Cornell Medicine under IRB-approved protocols (WCM 1410015560 and 1603017108). All patients provided written informed consent. Cells were isolated and maintained in DMEM, supplemented with 10% fetal calf serum (FCS), 100 U/ml penicillin/streptomycin and 1% sodium pyruvate (ThermoFisher Scientific). Cells were grown at 37°C in a humidified atmosphere with 5% CO2. See Materials and method Details for cell growth and proliferation work specific procedures.

### Human breast cancer cells

Human breast cancer cell lines, MCF7, T47D, ZR75, MDA-MB-231, MDA-MB-468, SKBR3, Hs578T and HEK293T, DU4475 and MDA-MB-157 were purchased from ATCC, USA. Human 4-OH tamoxifen resistant breast cancer cells (LCC2) were obtained from Prof. Robert Clarke. RKO, RKO-T29, RKO-A19, HCT116, HCT116-HWT, HCT116-HMUT were obtained from Dr. Jihye Yun and Dr. Bert Vogelstein and culture methods were performed as described previously (*Yun et al., 2009*). MCF7, MDA-MB-231, MDA-MB-468, MDA-MB-157, Hs578T, HEK293T were grown in DMEM (Invitrogen), supplemented with 10% fetal calf serum (FCS) (Invitrogen), 100 U/ml penicillin/streptomycin and 1% sodium pyruvate. ZR75, T47D, SKBR3 and DU4475 were grown in RPMI, supplemented with 10% FCS, 100 U/ml penicillin/streptomycin and 2 mM L-Glutamine (ThermoFisher Scientific). LCC2 were grown in phenol red-free DMEM, supplemented with 10% charcoal stripped serum (CCS; ThermoFisher Scientific), 100 U/ml penicillin/streptomycin and 1% sodium pyruvate. All cell lines were grown at 37°C with 5% $CO_2$ in a humidified environment. Mycoplasma detection of all cell lines were tested and negative results were observed. Authenticated cell lines were used within six months of purchase from the ATCC.

### Animal experimental model

All animal experiments were performed in Monash Animal Research Platform at Hudson Institute for Medical Research and Research Animal Resource Center at Weill Cornell Medicine. Experimental procedures were in accordance with guidelines for Animal Care and Use, approved by Monash University Animal Ethics Committee (MMCA2014/20) and Weill Cornell Medicine IACUC Protocol

(REQ00016929). Athymic Balb/c nude, FVB/N and NOD Scid gamma (NSG) mice were used for this study. All mouse procedures were performed with 6–8 weeks old female mice and treatments administered via subcutaneous injection. See Materials and Methods details for *in vivo* work specific procedures.

## Methods details

### Cell growth and proliferation assays

3D culture of a panel of breast cancer and colon cancer cells were seeded at a density of 3,000 cells per well in media containing 30% growth-factor-reduced matrigel or 5 mg/ml collagen in optical-bottom 96-well plates. Patient-derived breast tumors were first dissociated into single cell suspensions and seeded at a density of 32,000 cells per well in media containing 30% matrigel. Cells were serum-starved overnight and cultured under different experimental conditions: 10–1000 pM unacylated ghrelin, 100 pM ghrelin, $1.5 \times 10^{-13}$- $1.5 \times 10^{-6}$ M doxorubicin, $1 \times 10^{-18}$–$1 \times 10^{-7}$ M AZP-531, 20–200 ng/ml pertussis toxin, 10 µM U0126 (MEK inhibitor), 10 µM LY294002 (PI3K inhibitor), 60 nM KT5720, 1.4 µM SQ22536, 15 µM cAMPs-RP, 10 nM melatonin and the combination of doxorubicin/unacylated ghrelin or doxorubicin/AZP-531, with or without 10 nM estradiol or 10% serum or 1 µM forskolin and medium was replaced every 2 days. At the end of the experimental time point, cells were fixed in 100% methanol. The total number of cells, or the number of dead or proliferating cells, was assessed using Hoechst 33342, EarlyTox Dead Assay Kit, or Click-IT EdU Kit, respectively, and analyzed according to the manufacturer's protocols.

### Microscope image acquisition

Imaging of wells with greater than 95% coverage was acquired using a 10X/NA 0.8 objective by tiling $3 \times 3$ using confocal microscopy (Zeiss LSM880) with Axiocam. The fixed cells were imaged at room temperature, while the live cell images were acquired at 37°C placed inside a temperature and $CO_2$ regulated chamber. The number of nuclei/Ethidium Homodimer-III+/EdU+ cells were measured using the Zeiss Enhanced Navigation (Zen) and counted using the Imaris software.

### Binding assays

Cy3-tagged UAG binding assays were performed in 3D cultures of breast cancer cells. MCF7, MDA-MB-468 and MDA-MB-231 cells were seeded at a density of 3,000 cells per well in Matrigel in optical-bottom 96-well plates. Cells were serum-starved overnight. Hoechst nuclear stain was added prior to the live cell imaging. 1 µM Cy3-tagged UAG with 10% serum was then added and time-lapse confocal imaging was performed to examine the localization of peptide binding.

### BRAF transient transfections

Two million cells were harvested and transfected with or without 1.5 µg of BRAFV600E plasmid (obtained from Dr. Dan Gough) using AMAXA Nucleofector (Lonza), according to supplier's instructions. Transfected cells were then cultured according to above sections (cell growth and proliferation assay). After 5 days, cell number per field was assessed using Hoechst nuclear stain, confocal microscopy and Image J.

### GNAI CRISPR KO generation

The lentiviral construct, lentiCRISPRv2 (containing hSpCas9 and the chimeric guide RNA cassettes), was digested using BsmBI. Prior to ligation, each pair of oligos (100 µM) were annealed. The oligos (GNAI sequence guide strands) designed were based on the target site sequence (20 bp) and were flanked on the 3' end by a 3 bp NGG PAM sequence. See *Supplementary file 1* - Oligonucleotide Sequences for GNAI sequence guide strands used in this study.

The gel-purified, BsmBI digested plasmid was ligated to the diluted (1:200) annealed oligo. The ligation product was transformed into competent *Escherichia coli* Stbl3TM cells. Ampicillin resistance colonies were selected for miniprep/maxiprep.

For virus production, HEK293T cells were plated in a 10 cm dish and transfected 24 hr later (80% confluence) with a prepared mix in DMEM media (no supplements) containing 5 µg of empty vector or gRNA plasmid of GNAI1, GNAI2 or GNAI3, 2.5 µg of psPAX2, 1.25 µg of VSV.G, and 15 µl of

polyethylenimine (PEI; 1 µg/ml). 24 hr following transfection, media was replaced and supernatants (GNAI1, GNAI2 or GNAI3) were harvested and collected every 24 hr up to 72 hr post-transfection.

To generate CRISPR/Cas9 sgRNA stable breast cancer cell lines, breast cancer cells were plated in a 6-well plate. 24 hr following plating, cells were transfected with CRISPR/Cas9 sgRNA lentiviral vector and 8 µg/ml of polybrene. 24 hr after transfection, media was replaced. Cells were then selected in puromycin (2 µg/ml) for 5 days. CRISPR/Cas9 sgRNA stable breast cancer cell lines were then transduced with viral supernatants (GNAI1, GNAI2 or GNAI3 virus) in the presence of polybrene (8 µg/ml). 24 hr after transduction cells were selected in Blasticidin S (2 µg/ml) for 5 days. Selected cells were used to perform cell growth assays.

## ERK and Akt activity assays

ERK activity assay: The EKAR FRET-based system was used to monitor ERK activity. Briefly, breast cancer cells were co-transfected with pPBJ—puro-FRET3-EKAR-nls and the pCMV- hyPBase transposase vector (obtained from Dr. John Albeck), and stably transfected cells selected using FACS for EYFP and ECFP-positive cells. Cells were then cultured in 3D and serum-starved overnight. Prior live cell time-lapse confocal imaging, medium was replaced with 100 pM unacylated ghrelin and 10% serum. Images were acquired following FRET and analyzed using Imaris software. Data were normalized to ECFP signal.

AKT activity assay: pMSCV-puro-Foxo3a-H212R-N400-mCherry (obtained from Dr. John Albeck) was transfected into breast cancer cells using Amaxa Nucleofector. After 3 days of puromycin (2 µg/ml) selection, transfected cells were cultured in 3D. Cells were serum-starved overnight. Prior to live cell imaging, medium was replaced with 100 pM unacylated ghrelin and 10% serum. Time-lapse imaging was performed to examine mCherry-tagged FoxO3 localization. Data were analyzed using Hoechst nuclear stain to mask nuclei using Imaris software.

## cAMP assay

The Lance Ultra cAMP kit was used to measure the effect of unacylated ghrelin on cAMP production. All kit components were prepared according to manufacturer's specifications. Briefly, MCF7 (475 cells/well) were incubated at room temperature for 60 min with unacylated ghrelin (10–1000 pM), in the absence or presence of forskolin (0.5 µM). The Eu-cAMP tracer and Ulight-anti cAMP reagents were then added for 1 hr at room temperature. The plate was then read using the Envision TRF capable reader (Perkin Elmer) and fluorescence was measured with excitation wavelengths of 340 or 340 nm and emission of 665 nm according to the manufacturer's instructions.

## Intracellular calcium release assay

Intracellular $Ca^{2+}$ levels were measured in MCF7, ZR75 and J110 cells by fluorescence using the Flexstation (Molecular Devices, Sunnyvale, CA, USA) as previously described (*Callaghan et al., 2012*). Briefly, cells were plated, allowed to reach 50–70% confluency, and loaded with 2 µM fura 2-AM for 1 hr in the presence of 2.5 mM probenecid and 0.01% pluronic F-127 at 37°C. Cells were then washed twice with assay buffer, and changes in fluorescence in response to drug addition were measured over 100 s using excitation wavelengths of 340 and 380 nm and emission of 520 nm.

## Gαi activation assay

Gαi activation assay was performed on MCF7 cells according to the manufacturer's protocol. Briefly, cell lysates from 3D cell culture were incubated with an anti-active Gαi antibody (1:1000). The precipitated active Gαi was immunoblotted with an anti- Gαi antibody (1:1000). Bound antibodies were revealed with HRP-conjugated secondary antibodies (1:2000) using SuperSignal West pico chemiluminescent solution (Pierce, Rockford, IL). Protein amount was normalized to β–tubulin. Membranes were scanned and the densitometric analysis of the bands was performed using the ChemiDoc MP imaging system (BioRad).

## Western blot analysis

Western blotting was performed as described previously (*Brown et al., 2009*). Briefly, cells isolated from or 5 mg/ml Collagen (obtained from Dr. Jason Spector) were lysed in RIPA lysis buffer (Sigma-Aldrich) supplemented with 100x Protease/Phosphatase inhibitor cocktail (Cell Signaling Technology

Inc). Cell extracts (20 µg per lane) were separated by NuPAGE 4–12% Bis-Tris protein gels (Thermo-Fisher Scientific) and transferred to nitrocellulose membranes. The following primary antibodies were used: CDK4, Cyclin D3, phospho-Rb (Ser795), Rb, Bcl-2, Bax, phospho-p44/42 MAPK (Erk1/2) (Thr202/Tyr204), p44/42 MAPK (Erk1/2), phospho-p90RSK (Ser380), RSK1/RSK2/RSK3, c-Myc, phospho-Akt (Ser473), Akt, phospho-p70 S6 Kinase (Thr389), p70 S6 Kinase. Bound antibodies were revealed with HRP conjugated secondary antibodies. The following secondary antibodies were used: Goat anti-mouse IgG H and L (HRP), Donkey anti- Rabbit IgG H and L (HRP). Detected bands were consistent with expected molecular weights based on antibody datasheets (Key Resource Table). β–actin was used as a loading control. See Key Resources Table for antibody details used in this study. Membranes were scanned using the western lightning plus-ECL (Thermo Fisher Scientific). Signal intensities were quantified using ImageLab software (BioRad).

## Fluorescence activated cell sorting (FACS)

FACS analysis was performed to characterize effects of unacylated ghrelin on cell cycle and apoptosis. Cells were plated at a density of 500,000 cells in a 10 $cm^2$ petri dish. Cells were serum- starved overnight with phenol red-free media and treated with different concentrations (0–100 pM) of unacylated ghrelin, with or without 10 nM estradiol. In order to determine effects on cell cycle, cells were treated for 5 days, with media being changed every 2 days. After 5 days, cells were harvested and fixed with ice cold 70% ethanol, stored overnight at −20°C, and stained with propidium iodine (PI) staining buffer (1 mg/ml RNase A, 0.1% Triton X-100, 100 µg/ml PI in PBS). To evaluate effects on cell apoptosis, cells were treated for 6 hr, harvested and then stained with Annexin V-FITC for 15 min at room temperature in PBS. Cells were analysed with a FACSCANTO II flow cytometer (BD Biosciences, USA). For FACS data analysis by FlowJo software, forward scatter (FS) vs. side scatter (SS) plots were used for gating cells and to identify any changes in the scatter properties of the cells. Annexin V FITC-A vs Propidium Iodide-A plots from the gated cells show the populations corresponding to viable and non-apoptotic (Annexin V–PI–), early (Annexin V+PI–), and late (Annexin V+PI+) apoptotic cells.

## Analysis of cell cycle progression using the fluorescence ubiquitination cell cycle indicator (FUCCI)

To investigate cell cycle progression and division in live cells, we used the fluorescent ubiquitination-based cell cycle indicator (FUCCI) to track the G1/G0 phase and S/G2/M phases. Breast cancer cells (15 × $10^4$ cells/well) in six wells plate were seeded in 2D and transduced with the Premo FUCCI Cell Cycle Sensor according to the manufacturer's protocol. After optimal expression of the FUCCI sensor was achieved (16 hr), cells were detached by TrypLE Express (Thermo Fisher Scientific), counted using a hemocytometer and seeded in media containing 30% matrigel. Cells were serum-starved overnight, treated with 100 pM unacylated ghrelin for 5 days with 10% serum and medium was replaced every 2 days. At endpoint, fluorescence was analyzed using confocal microscopy and cell counts obtained. Data are presented as a percentage of the total number of fluorescent cells examined.

## Xenograft and allograft studies

One million MCF7 or ZR75 cells were injected into the mammary fat pad of 6-week-old ovary-intact female athymic Balb/c nude (immunodeficient) mice. Estrogen pellets were prepared in the laboratory using 17β-estradiol (estrogen) powder and silicone according to previous publications (*Laidlaw et al., 1995*) and were implanted (0.8 mg/pellet, 60-day release) subcutaneously between the shoulder blades at the time of ER+ breast cancer cell injection. After injection, the mice were monitored daily for well-being, pain and distress, and for tumor growth by palpation. Tumors typically appeared within 14 days, with an engraftment rate of approximately 75%. Once palpable, the tumors were measured daily in the long (L) and short (W) axes with digital callipers, and tumor volume estimated using the standard formula (LxW$^2$)/2. Once the tumor reached a volume of 200 $mm^3$, the animal was randomized to receive saline (vehicle control) or unacylated ghrelin) via subcutaneous injection for a maximum of 28 days or until tumor size reached 10 mm in any axis, at which point the mice were humanely sacrificed by cervical dislocation.

A syngeneic breast cancer mouse model was created in female FVB/N mice. $1.25 \times 10^5$ J110 cells (obtained from Dr. Myles Brown) were injected into the mammary fat pad of 6-week-old ovary-intact FVB (Immunocompetent) mice. Once the tumor reached a volume of 70 mm$^3$, the mice were randomized to receive saline (vehicle control) or treatment (unacylated ghrelin or AZP-531) via subcutaneous injection. After 7 days of treatment, the mice were humanely sacrificed by cervical dislocation. Additional studies were also performed in NSG mice. MDA-MB-468 ($1 \times 10^6$) were injected into the mammary fat pad of 6 week old ovary-intact NSG (immunodeficient) mice. Once the tumor reached a volume of 150 mm$^3$, the mice were randomized to receive saline (vehicle control) or AZP-531 via subcutaneous injection. After 28 days of treatment, the mice were humanely sacrificed by cervical dislocation.

## Measurement of apoptotic cells from xenograft/allograft studies

MCF7, ZR75 and J110 tumor sections post treatment with 100 µg/kg or 200 µg/kg of unacylated ghrelin were assessed using Hoechst nuclear stain and confocal microscopy. Quantification of cells with apoptotic nuclei was performed using Image J (Fiji).

## Microarray datasets

Gene expression profiles from GSE34412 were retrieved via GEO2R (https://www.ncbi.nlm.nih.gov/geo/geo2r/). Expression profiles were based on a Custom Human Agilent array (GPL8269). GEO2R was used to perform differential gene expression analysis between responders (GSM847888, GSM847905) and non-responders (GSM847901, GSM847893). Ingenuity Pathway Analysis (IPA) was used to determine regulatory pathways that distinguished between responders and non-responders. Differences in gene expression were consistent with activation of upstream regulator MAPK1 in non-responders vs. responders (Z-score 3.888; overlap p-value 3.07E-15), with 86 genes of 144 having measurements consistent with MAPK1 activation. Log expression values corresponding to the MAPK target genes were extracted from GEO2R for each sample. If a gene was represented by more than one probe, the expression value corresponding to the probe with the largest absolute fold-change between responders and non-responders was selected. Genes predicted by IPA to be activated were visualized in a heatmap (expression values were centered and scaled).

## Statistical analysis

All experiments were performed at least twice with n $\geq$ 3 per experiment and all data are expressed as the mean $\pm$ SEM. Statistical analysis was carried out with software Graph Pad Prism 7. For experiments with multiple comparisons, statistical analysis was performed using one-way ANOVA followed by Dunnett multiple comparison, where means of each column were compared to the mean of a control column. For experiments with comparison of two independent groups, statistical analysis was performed using two-way ANOVA followed by Dunnett multiple comparison, where means of each experimental condition were compared to the control. A p-value was reported and significance was classified as p<0.05(*), p$\leq$0.005(**), p$\leq$0.0005(***), p$\leq$0.0001(****), n.s.: not significant.

## Acknowledgements

We thank R Clarke, J Yun, B Vogelstein, M Brown for providing cell lines. We are grateful to D Gough (BRAF transient transfections), J Albeck (EKAR and AKT activity assay) and M Foronda (CRISPR) for provision of reagents and valuable experimental advice. We acknowledge S Yomtoubian for providing information relating to TNBC subtype and T Pham for assistance with data analysis. We are indebted J Cain, SN Jayasekara and M Docanto for assistance with in vivo work; the Monash Micro Imaging Facility, Monash Animal Research Platform, MHTP Medical Genomics Facility, MHTP Flow Cytometry Facility and Research Animal Resource Center at Weill Cornell Medicine for valuable discussions and comments.

# Additional information

## Competing interests

Kristy A Brown: Cornell University has filed a patent application on the work that is described in this paper with Dr Brown as a named inventor. The other authors declare that no competing interests exist.

## Funding

| Funder | Grant reference number | Author |
|---|---|---|
| National Breast Cancer Foundation | NC-14-011 | Kristy A Brown |
| State Government of Victoria | | Kristy A Brown |
| National Health and Medical Research Council | GNT1007714 | Kristy A Brown |
| The Endocrine Society of Australia | | Kristy A Brown |
| National Breast Cancer Foundation | ECF-16-004 | Kristy A Brown |
| National Breast Cancer Foundation | | Kara Britt |

The funders had no role in study design, data collection and interpretation, or the decision to submit the work for publication.

## Author contributions

CheukMan C Au, Conceptualization, Data curation, Formal analysis, Funding acquisition, Validation, Investigation, Methodology, Writing - original draft, Writing - review and editing; John B Furness, Xin Yun Huang, Supervision; Kara Britt, Investigation, Methodology; Sofya Oshchepkova, Heta Ladumor, Investigation; Kai Ying Soo, Formal analysis, Investigation, Methodology; Brid Callaghan, Celine Gerard, Yufeng Wang, Formal analysis, Investigation; Giorgio Inghirami, Vivek Mittal, Eleni Andreopoulou, Resources; Jason A Spector, Resources, Methodology; Paul Zumbo, Formal analysis, Visualization; Doron Betel, Formal analysis, Supervision; Lukas Dow, Methodology; Kristy A Brown, Conceptualization, Resources, Formal analysis, Supervision, Funding acquisition, Methodology, Writing - original draft, Project administration, Writing - review and editing

## Author ORCIDs

Kristy A Brown (iD) https://orcid.org/0000-0003-3382-5546

## Ethics

Human subjects: Patient-derived breast tumors (PDX (X. Zhang et al., 2013)) were obtained from Dr. Giorgio Inghirami and estrogen receptor positive (ER+) breast tumors were obtained from Dr. Eleni Andreopoulou at Weill Cornell Medicine under IRB-approved protocols (WCM 1410015560 and 1603017108). All patients provided written informed consent.

Animal experimentation: All animal experiments were performed in Monash Animal Research Platform at Hudson Institute for Medical Research and Research Animal Resource Center at Weill Cornell Medicine. Experimental procedures were in accordance with guidelines for Animal Care and Use, approved by Monash University Animal Ethics Committee (MMCA2014/20) and Weill Cornell Medicine IACUC Protocol (REQ00016929).

## Decision letter and Author response

Decision letter https://doi.org/10.7554/eLife.56913.sa1
Author response https://doi.org/10.7554/eLife.56913.sa2

## Additional files

### Supplementary files
- Supplementary file 1. Oligonucleotide Sequences.
- Transparent reporting form

### Data availability

All data generated or analysed during this study are included in the manuscript and supporting files. Source data files have been provided for all figures.

The following previously published dataset was used:

| Author(s) | Year | Dataset title | Dataset URL | Database and Identifier |
|---|---|---|---|---|
| Zhang X, Prat A, Perou CM | 2013 | A Renewable Tissue Resource of Phenotypically Stable, Biologically and Ethnically Diverse, Patient-derived Human Breast Cancer Xenografts | https://www.ncbi.nlm.nih.gov/geo/query/acc.cgi?acc=GSE34412 | NCBI Gene Expression Omnibus, GSE34412 |

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

# Appendix 1

**Appendix 1—key resources table**

| Reagent type (species) or resource | Designation | Source or reference | Identifiers | Additional information |
|---|---|---|---|---|
| Strain, strain background (*Escherichia coli*) | One Shot Stbl3 Chemically Competent *E. coli* | ThermoFisher Scientific | Cat# C737303 | |
| Strain, strain background (*Mus-musculus*) | BALB/c-Foxn1$^{nu}$/Arc (BALB/c nude) mice | Animal Resources Centre, Australia | Cat# BCNU; RRID:MGI:2161064 | Female, 6 week old |
| Strain, strain background (*Mus-musculus*) | FVB/NJArc (FVB/N) mice | Animal Resources Centre, Australia | Cat# 001800; RRID:IMSR_JAX:001800 | Female, 6 week old |
| Strain, strain background (*Mus-musculus*) | NOD.Cg-Prkdc$^{scid}$Il2rg$^{tm1Wjl}$/SzJ (NSG) mice | NOD Scid gamma, Jackson laboratory | Cat# 005557; RRID:IMSR_JAX:005557 | Female, 6 week old |
| Cell line (*Homo-sapiens*) | MCF7 | ATCC | ATCC Cat# HTB-22; RRID:CVCL_0031 | |
| Cell line (*Homo-sapiens*) | T47D | ATCC | ATCC Cat# HTB-133; RRID:CVCL_0553 | |
| Cell line (*Homo-sapiens*) | ZR75-1 | ATCC | ATCC Cat# CRL-1500; RRID:CVCL_0588 | |
| Cell line (*Homo-sapiens*) | MDA-MB-231 | ATCC | ATCC Cat# HTB-26; RRID:CVCL_0062 | |
| Cell line (*Homo-sapiens*) | MDA-MB-468 | ATCC | ATCC Cat# HTB-132; RRID:CVCL_0419 | |
| Cell line (*Homo-sapiens*) | SKBR3 | ATCC | ATCC Cat# HTB-30; RRID:CVCL_0033 | |
| Cell line (*Homo-sapiens*) | Hs578T | ATCC | ATCC Cat# HTB-126; RRID:CVCL_0332 | |
| Cell line (*Homo-sapiens*) | HEK293T | ATCC | ATCC Cat# CRL-1573; RRID:CVCL_0045 | |
| Cell line (*Homo-sapiens*) | DU4475 | ATCC | ATCC Cat# HTB-123; RRID:CVCL_1183 | |

*Appendix 1—key resources table continued*

| Reagent type (species) or resource | Designation | Source or reference | Identifiers | Additional information |
|---|---|---|---|---|
| Cell line (*Homo-sapiens*) | MDA-MB-157 | ATCC | ATCC Cat# HTB-24; RRID:CVCL_0618 | |
| Cell line (*Homo-sapiens*) | LCC2 | obtained from Prof. Robert Clarke | | |
| Cell line (*Homo-sapiens*) | RKO | *Yun et al., 2009* | | |
| Cell line (*Homo-sapiens*) | RKO-T29 | *Yun et al., 2009* | | |
| Cell line (*Homo-sapiens*) | RKO-A19 | *Yun et al., 2009* | | |
| Cell line (*Homo-sapiens*) | HCT116 | *Yun et al., 2009* | | |
| Cell line (*Homo-sapiens*) | HCT116-HWT | *Yun et al., 2009* | | |
| Cell line (*Homo-sapiens*) | HCT116-HMUT | *Yun et al., 2009* | | |
| Cell line (*Homo-sapiens*) | J110 | obtained from Dr. Myles Brown | | |
| Biological sample (human) | Patient-derived breast tumors | obtained from Dr. Giorgio Inghirami | | See Materials and methods section |
| Biological sample (human) | Estrogen receptor positive (ER +) breast tumors | obtained from Dr. Eleni Andreopoulou | | See Materials and methods section |
| Antibody | Goat anti-mouse IgG H and L (HRP) | Abcam | Cat# ab205719; RRID:AB_2755049 | 1:5000 |
| Antibody | Donkey anti-Rabbit IgG H and L (HRP) | Abcam | Cat# ab7083; RRID:AB_955416 | 1:5000 |
| Antibody | Mouse Monoclonal anti-Tubulin, beta, (KMX-1) | MilliporeSigma | Cat# MAB3408; RRID:AB_94650 | 1:10000 |
| Antibody | Mouse Monoclonal anti-$\beta$-Actin—Peroxidase | Sigma-Aldrich | Cat#A3854; RRID:AB_262011 | 1:5000 |
| Antibody | Rabbit Monoclonal anti-CDK4 (D9G3E) | Cell Signaling | Cat# 12790; RRID:AB_2631166 | 1:1000 |
| Antibody | Mouse Monoclonal anti-Cyclin D3 (DCS22) | Cell Signaling | Cat# 2936; RRID:AB_2070801 | 1:2000 |

*Appendix 1—key resources table continued*

| Reagent type (species) or resource | Designation | Source or reference | Identifiers | Additional information |
|---|---|---|---|---|
| Antibody | Rabbit Polyclonal anti-phospho-Rb (Ser795) | Cell Signaling | Cat# 9301; RRID:AB_330013) | 1:1000 |
| Antibody | Mouse Monoclonal anti-Rb (4H1) | Cell Signaling | Cat# 9309; RRID:AB_823629 | 1:2000 |
| Antibody | Rabbit Monoclonal anti-Bcl-2 (D55G8) (Human Specific) | Cell Signaling | Cat# 4223; RRID:AB_1903909 | 1:1000 |
| Antibody | Rabbit Monoclonal anti-Bax | Cell Signaling | Cat# 2772; RRID:AB_10695870 | 1:1000 |
| Antibody | Rabbit Polyclonal anti-phospho-p44/42 MAPK (Erk1/2) (Thr202/Tyr204) | Cell Signaling | Cat# 9101; RRID:AB_331646 | 1:1000 |
| Antibody | Rabbit Monoclonal anti-p44/42 MAPK (Erk1/2) (137F5) | Cell Signaling | Cat# 4695; RRID:AB_390779 | 1:1000 |
| Antibody | Rabbit Monoclonal anti-phospho-p90RSK (Ser380) (D3H11) | Cell Signaling | Cat# 11989; RRID:AB_2687613 | 1:1000 |
| Antibody | Rabbit Monoclonal anti-RSK1/RSK2/RSK3 (32D7) | Cell Signaling | Cat# 9355; RRID:AB_659900 | 1:1000 |
| Antibody | Rabbit Monoclonal anti-c-Myc (D84C12) | Cell Signaling | Cat# 5605; RRID:AB_1903938 | 1:1000 |
| Antibody | Rabbit Polyclonal anti-phospho-Akt (Ser473) | Cell Signaling | Cat# 9271; RRID:AB_329825 | 1:1000 |
| Antibody | Rabbit Polyclonal anti-Akt | Cell Signaling | Cat# 9272; RRID:AB_329827 | 1:1000 |
| Antibody | Rabbit Monoclonal anti-phospho-p70 S6 Kinase (Thr389) (108D2) | Cell Signaling | Cat# 9234; RRID:AB_2269803 | 1:1000 |
| Antibody | Rabbit Monoclonal anti-p70 S6 Kinase (49D7) | Cell Signaling | Cat# 2708; RRID:AB_390722 | 1:1000 |
| Recombinant DNA reagent | pPBJ—puro-FRET3-EKAR-nls | obtained from Dr. John Albeck | | |
| Recombinant DNA reagent | pCMV-hyPBase transposase vector | obtained from Dr. John Albeck | | |
| Recombinant DNA reagent | pMSCV-puro-Foxo3a-H212R-N400-mCherry | obtained from Dr. John Albeck | | |
| Recombinant DNA reagent | lentiCRISPR v2 | *Sanjana et al., 2014* | Addgene Cat# 52961; RRID:Addgene_52961 | |
| Recombinant DNA reagent | BRAFV600E plasmid | obtained from Dr. Dan Gough | | |

*Appendix 1—key resources table continued*

| Reagent type (species) or resource | Designation | Source or reference | Identifiers | Additional information |
|---|---|---|---|---|
| Recombinant DNA reagent | psPAX2 | This paper | Addgene Cat# 12260; RRID:Addgene_12260 | |
| Recombinant DNA reagent | VSV-G | *Reya et al., 2003* | Addgene Cat# 14888; RRID:Addgene_14888 | |
| Sequence-based reagent | See *Supplementary file 1* for GNAI sequence guide strands used in this study | | | |
| Peptide, recombinant protein | Rat des-octanoyl ghrelin | China Peptides | Cat# Rat des-octanoyl ghrelin | |
| Peptide, recombinant protein | [Des-octanoyl]-Ghrelin (rat) | Tocris Bioscience | Cat# 2951 | |
| Peptide, recombinant protein | Ghrelin (rat) | Tocris Bioscience | Cat# 1465 | |
| Peptide, recombinant protein | AZP531 | MedChem Express | Cat# HY-P0231 | |
| Peptide, recombinant protein | Cy3-tagged UAG | Pepmic Co, LtD | Cat# Cy3-GR-28 | |
| Commercial assay or kit | EarlyTox Live/Dead Assay Kit | Molecular Devices | Cat# P/N R8340 | |
| Commercial assay or kit | Click-iT Plus EdU Alexa Fluor 488 Imaging Kit | ThermoFisher Scientific | Cat# C10637 | |
| Commercial assay or kit | Cell Line NucleofectorKit V | Lonza | Cat# VCA-1003 | |
| Commercial assay or kit | Lance Ultra cAMP Detection Kit | Perkin Elmer | Cat# TRF0262 | |
| Commercial assay or kit | G$\alpha_i$ Activation Assay Kit | New East Biosciences | Cat# 80301 | |
| Commercial assay or kit | Premo FUCCI Cell Cycle Sensor (BacMam 2.0) | ThermoFisher Scientific | Cat# P36237 | |
| Commercial assay or kit | PureYield Plasmid Miniprep System | Promega | Cat# A1223 | |
| Commercial assay or kit | PureYield Plasmid Maxiprep System | Promega | Cat# A2392 | |
| Chemical compound, drug | Pertussis Toxin from *B. pertussis*, Lyophilized (Salt-Free) | List Biological Laboratories | Cat# 181 | |
| Chemical compound, drug | U0126 | Cell Signaling | Cat# 9903S | |
| Chemical compound, drug | LY294002 | Cell Signaling | Cat# 9901 | |

*Appendix 1—key resources table continued*

| Reagent type (species) or resource | Designation | Source or reference | Identifiers | Additional information |
|---|---|---|---|---|
| Chemical compound, drug | KT 5720 | Tocris Bioscience | Cat# 1288 | |
| Chemical compound, drug | SQ 22536 | Tocris Bioscience | Cat# 1435 | |
| Chemical compound, drug | cAMPS-Rp, triethylammonium salt | Tocris Bioscience | Cat# 1337 | |
| Chemical compound, drug | Melatonin | Tocris Bioscience | Cat# 3550 | |
| Chemical compound, drug | Doxorubicin hydrochloride | Sigma-Aldrich | Cat# D1515 | |
| Chemical compound, drug | Forskolin | Sigma-Aldrich | Cat# F3917 | |
| Chemical compound, drug | $\beta$-Estradiol | Sigma-Aldrich | Cat# E8875-1G | |
| Software, algorithm | Prism | GraphPad | RRID:SCR_005375 | http://www.graph-pad.com/scientific-software/prism/ |
| Software, algorithm | Imaris | Bitplane | RRID:SCR_007370 | https://imaris.oxinst.com/ |
| Software, algorithm | Fiji | ImageJ | RRID:SCR_002285 | https://imagej.net/Fiji |
| Software, algorithm | Image Lab | BIO-RAD | RRID:SCR_014210 | http://www.biorad.com/en-us/product/image-lab-software?ID=KRE6P5E8Z; |
| Software, algorithm | ZEN software | ZEISS | RRID:SCR_013672 | https://www.zeiss.com/microscopy/us/products/microscope-software/zen-lite.html |
| Software, algorithm | Ingenuity Pathway Analysis | Qiagen | RRID:SCR_008653 | https://www.qiagen-bioinformatics.com/products/ingenuity-pathway-analysis |
| Software, algorithm | FlowJo software | FlowJo | RRID:SCR_008520 | https://www.flowjo.com/solutions/flowjo |
| Other | Hoechst 33342, Trihydrochloride, Trihydrate | ThermoFisher Scientific | Cat# H3570 | |
| Other | FITC Annexin V | BD Pharmingen | Cat# 556420 | |
| Other | Propidium Iodine | ThermoFisher Scientific | Cat# P1304MP | |
| Other | Fura-2, AM, cell permeant | ThermoFisher Scientific | Cat# F1221 | |
| Other | Probenecid, Water Soluble | ThermoFisher Scientific | Cat# P36400 | |

*Appendix 1—key resources table continued*

| Reagent type (species) or resource | Designation | Source or reference | Identifiers | Additional information |
|---|---|---|---|---|
| Other | Pluronic F-127 | ThermoFisher Scientific | Cat# P6867 | |
| Other | BsmBI | New England BioLabs | Cat# R0580S | |
| Other | Polyethylenimine, linear (PEI) | Polysciences, Inc | Cat# 23966–1 | |
| Other | Polybrene | Santa Cruz Biotechnology | Cat# sc-134220 | |
| Other | Puromycin Dihydrochloride | ThermoFisher Scientific | Cat# A1113803 | |
| Other | Blasticidin S HCl (10 mg/mL) | ThermoFisher Scientific | Cat# A1113903 | |
| Other | Corning Matrigel Growth Factor Reduced (GFR) Basement Membrane Matrix, Phenol Red-free, *LDEV-free, | Corning | Cat# 356231 | |
| Other | Collagen | Obtained from Dr. Jason Spector | | |

