## [Decision Letter]

**Acceptance summary:**

The hormone ghrelin is responsible for the stimulation of growth hormone release and appetite, yet little is known of the effects of its unacylated form. The present study uncovers an inhibitory role of unacylated ghrelin on breast cancer cells, and explores in vivo and ex vivo efficacy of a therapeutic analogue AZP-531. Targeting unacylated ghrelin may provide a new approach for the breast cancer treatment.

**Decision letter after peer review:**

Thank you for submitting your article "Three-dimensional growth of breast cancer cells potentiates the anti-tumor effects of unacylated ghrelin and AZP-531" for consideration by *eLife*. Your article has been reviewed by three peer reviewers, including Yi Arial Zeng as the Reviewing Editor and Reviewer #1, and the evaluation has been overseen Richard White as the Senior Editor. The following individuals involved in review of your submission have agreed to reveal their identity: Jason Carroll (Reviewer #2); Jeremy Borniger (Reviewer #3).

The reviewers have discussed the reviews with one another and the Reviewing Editor has drafted this decision to help you prepare a revised submission.

In particular, please pay attention to the comments 1-2 by reviewer 1 and comments 1-3 by reviewer 2. While some of these will require additional data, others may be addressed be a careful editing of the conclusions made in the manuscript.

Reviewer #1:

The manuscript by Au et al., by the Brown lab uncovers a new role of unacylated ghrelin (UAG) for breast cancer cell inhibition. This inhibition effect is obvious in 3D culture, but not in 2D culture that was used in previous studies. Au and co-authors discover that UAG leads to cell cycle arrest and apoptosis, with a mechanism involves activation of Gαi and suppression of MAPK signaling. This mechanism is reflected by the lack of response of UAG when treating cancer cells with BRAF or HRAS mutation. A cyclic analog of UAG, AZP-531 also inhibits breast cancer cell growth in vitro and in xenografts.

Overall, this study is nicely designed and executed with a battery of in vitro and in vivo analyses. This study has a clinical implication for breast cancer treatment.

My only main concern is the clarity in writing and statistical analysis. They are specified below. Fortunately, those are issues that can be addressed without further experiments.

1) The statements regarding the effects on TNBC are confusing. In Figure 1C, the two UAG-responsive TNBC lines were MDA-MB-468 (basal) and MDA-MB-157 (mesenchymal), the three not-responsive TNBC lines were DU4474 (BRAF*), Hs578T (HRAS*) and MDA-MB-231 (BRAF* HRAS*). It is clear that the response is dependent on the mutation state rather than the basal/mesenchymal subtype.

In Figure 5G, UAG was tested in two TNBC-PDX lines. One responded, one didn't. One happened to be basal subtype, and the other is mesenchymal. BRAF and KRAS mutation had not been characterized in these patient samples, but MAPK-target genes were differentially expressed between responsive and no-responsive TNBC samples (subsection “Unacylated ghrelin and cyclic analog AZP-531 inhibit tumor growth in xenograft models and patient-derived tumor cells”). The statement that "UAG caused a significant reduction in the basal-like subset of TNBC cells, but not mesenchymal TNBCs" (subsection “Unacylated ghrelin and cyclic analog AZP-531 inhibit tumor growth in xenograft models and patient-derived tumor cells”) is misleading, so does the labeling of basal and mesenchymal in the figure. These should be fixed, probably by directly using the name of the PDX line.

2) The statistic analyses need to be clarified. In almost all panels, the "not significant (n.s)" has been missed. These should all be fixed. (Figure 1C-F, 2D, 2G, Figure 1—figure supplement 1A-C etc.)

There are several places where the n.s is especially important. For example, in Figure 1C, the n.s should be indicated for the three UAG-not-responsive TNBC cell lines. This is essential as "not-responsive" is one of the main conclusions in this panel. Similarly, in Figure 4F and Figure 5G, the n.s should be labelled for the UAG-not-responsive TNBC lines, MB-231 (4F) and PDX line 3204-TG6 (5G).

3) In many places of the figure legend, it went "…n=3. Experiments were repeated at least twice." Is n=3 here a technical repeat? If it is experimental repeat, it should be written as n>=2. Please modify or clarify.

Reviewer #2:

This manuscript investigates the role of the hormone ghrelin and the unacylated form of ghrelin, which has a distinct functional profile. The authors assess the growth inhibitory effects of unacylated ghrelin and ghrelin in a panel of breast cancer cells grown in 3D or 2D, revealing a distinction in the anti-tumor activity in specific models. Unacylated ghrelin was shown to only elicit growth suppressive activity in full media and some work is conducted to show that MCF7 response is blunted in the presence of B-Raf mutants. Some mechanistic work is provided to show that G-alpha-i is a potential mediator of unacylated ghrelin activity in a process that involves cAMP. Additional work is provided to link unacylated ghrelin with the MAPK pathway and some downstream endpoints are explored, with follow-up work showing that unacylated ghrelin mediates its effects via both cell cycle arrest and induced apoptosis. Finally, in vivo and ex vivo efficacy of unacylated ghrelin and a therapeutic analogue called AZP-531 is explored.

This is an interesting paper, on a very niche and underexplored hormonal pathway. The paper is well written and the figures are clear and convincing. Overall, the work is well conducted and the conclusions are valid, but a few issues need to be addressed.

- I understand the rationale behind expressing B-Raf mutants in MCF7 cells to assess the altered response to unacylated ghrelin, but what are the basal physiological effects of this on the MCF7 model? Does it change estrogen dependence or basal cell viability/phenotype? Given that B-Raf mutations are not associated with ER+ disease, forcing a non-relevant mutant into a cell line model, likely has profound basal changes that will confound the experimental questions.

- In the in vivo experiments, there was no mention of weight loss or potential physiological consequences of giving unacylated ghrelin (or analogues) on normal tissue functioning. Did the authors assess some of the standard 'health' indicators in this experiment?

- The use of AZP-531 is justified because of improved chemical properties, but the assumption is that all the previous pathway work (on cAMP, MAPK etc) are valid for AZP-531. Is this the case or is there any evidence to support the conclusions that the mechanistic insight derived from unacylated ghrelin is directly applicable to the mode of action for AZP-531? it would be useful if even just one or two of the key functional endpoints explored with unacylated ghrelin, were validated with AZP-531.

- The start of the Results section was confusing to a non-expert. The text states "The dependence on 3D culture was examined for unacylated ghrelin, where treatment with 100 pM resulted in suppression of cell growth in matrigel and collagen, but not in 2D (Figure 1—figure supplement 1A and 1B). At this dose, no effects on MDA-MB-231 cell growth were observed in either 2D or 3D cultures". It's not clear what cell model the first sentence refers to. I assumed it was the MDA-MB-231 cells, but the first line suggests there is growth inhibition in 3D, but the second sentence suggested there isn't growth inhibition in this cell line model.

Reviewer #3:

The authors examined the effects of unacylated ghrelin (UAG) on breast cancer cell growth in vitro and in xenograft models. This is an important area given the limited amount of research that has been done on the mechanisms of UAG actions on cancer cells. I applaud the authors for systematically characterizing the effects of UAG on cancer cell growth using 3D growth assays as well as many different cell lines with well characterized mutational signatures. Step-by-step inhibitor studies on different components of the cAMP, MAPK, and Akt signaling pathways strongly supports their hypothesis that UAG actions are mediated (at least in part) via a GPCR coupled to Gai. For each of the conclusions presented in the manuscript, several parallel experiments were conducted and for the most part, all of these acted synergistically to support the conclusions drawn from the studies. The paper is also well written and I did not have a problem following the narrative flow. I have no major concerns for the manuscript at this time.

---

## [Author Response]

Reviewer #1:The manuscript by Au et al., by the Brown lab uncovers a new role of unacylated ghrelin (UAG) for breast cancer cell inhibition. This inhibition effect is obvious in 3D culture, but not in 2D culture that was used in previous studies. Au and co-authors discover that UAG leads to cell cycle arrest and apoptosis, with a mechanism involves activation of Gαi and suppression of MAPK signaling. This mechanism is reflected by the lack of response of UAG when treating cancer cells with BRAF or HRAS mutation. A cyclic analog of UAG, AZP-531 also inhibits breast cancer cell growth in vitro and in xenografts.Overall, this study is nicely designed and executed with a battery of in vitro and in vivo analyses. This study has a clinical implication for breast cancer treatment.My only main concern is the clarity in writing and statistical analysis. They are specified below. Fortunately, those are issues that can be addressed without further experiments.1) The statements regarding the effects on TNBC are confusing. In Figure 1C, the two UAG-responsive TNBC lines were MDA-MB-468 (basal) and MDA-MB-157 (mesenchymal), the three not-responsive TNBC lines were DU4474 (BRAF*), Hs578T (HRAS*) and MDA-MB-231 (BRAF* HRAS*). It is clear that the response is dependent on the mutation state rather than the basal/mesenchymal subtype.In Figure 5G, UAG was tested in two TNBC-PDX lines. One responded, one didn't. One happened to be basal subtype, and the other is mesenchymal. BRAF and KRAS mutation had not been characterized in these patient samples, but MAPK-target genes were differentially expressed between responsive and no-responsive TNBC samples (subsection “Unacylated ghrelin and cyclic analog AZP-531 inhibit tumor growth in xenograft models and patient-derived tumor cells”). The statement that "UAG caused a significant reduction in the basal-like subset of TNBC cells, but not mesenchymal TNBCs" (subsection “Unacylated ghrelin and cyclic analog AZP-531 inhibit tumor growth in xenograft models and patient-derived tumor cells”) is misleading, so does the labeling of basal and mesenchymal in the figure. These should be fixed, probably by directly using the name of the PDX line.

We appreciate the reviewer’s insights and have modified the text and figures related to TNBC-PDX lines without focusing on their TNBC classification. Specifically, we have removed “basal” and “mesenchymal” labels from Figure 5G and Figure 4—figure supplement 1E, and associated figure legends. We have also modified the Results section as follows:

Subsection “Unacylated ghrelin and cyclic analog AZP-531 inhibits tumor growth in xenograft models and patient-derived tumor cells”: “Unacylated ghrelin at 100pM caused a significant reduction in the serum-induced growth of ER+ breast cancer cells and the basal-like subset of TNBC cells, but not mesenchymal TNBCs (Figure 5G, Figure 4—figure supplement 1E; Table 1).” was replaced with “Unacylated ghrelin at 100pM caused a significant reduction in the serum-induced growth of ER+ breast cancer cells, while inhibiting or having no effect in the TNBC patient samples examined (Figure 5G, Figure 4—figure supplement 1E; Table 1).”

2) The statistic analyses need to be clarified. In almost all panels, the "not significant (n.s)" has been missed. These should all be fixed. (Figure 1C-F, 2D, 2G, Figure 1—figure supplement 1A-C etc.)There are several places where the n.s is especially important. For example, in Figure 1C, the n.s should be indicated for the three UAG-not-responsive TNBC cell lines. This is essential as "not-responsive" is one of the main conclusions in this panel. Similarly, in Figure 4F and Figure 5G, the n.s should be labelled for the UAG-not-responsive TNBC lines, MB-231 (4F) and PDX line 3204-TG6 (5G).

We agree with the reviewer that addition of “n.s.” to bar graphs would emphasize a lack of effect of treatment and have modified Figure 1C-F, Figure 2C,D,G, Figure 4F, Figure 5D,G, Figure 1—figure supplement 1A-C, Figure 2—figure supplement 1B, Figure 3—figure supplement 1C,D,F,G,H accordingly. The definition of “n.s.” was added to the Statistical analysis section of the manuscript.

3) In many places of the figure legend, it went "…n=3. Experiments were repeated at least twice." Is n=3 here a technical repeat? If it is experimental repeat, it should be written as n>=2. Please modify or clarify.

In order to contend with routinely used in vitro experimental approaches without misleading reviewers, we replaced “n=x/group” with “x replicates/group” in all the relevant figure legends.

Reviewer #2:This manuscript investigates the role of the hormone ghrelin and the unacylated form of ghrelin, which has a distinct functional profile. The authors assess the growth inhibitory effects of unacylated ghrelin and ghrelin in a panel of breast cancer cells grown in 3D or 2D, revealing a distinction in the anti-tumor activity in specific models. Unacylated ghrelin was shown to only elicit growth suppressive activity in full media and some work is conducted to show that MCF7 response is blunted in the presence of B-Raf mutants. Some mechanistic work is provided to show that G-alphaα-i is a potential mediator of unacylated ghrelin activity in a process that involves cAMP. Additional work is provided to link unacylated ghrelin with the MAPK pathway and some downstream endpoints are explored, with follow-up work showing that unacylated ghrelin mediates its effects via both cell cycle arrest and induced apoptosis. Finally, in vivo and ex vivo efficacy of unacylated ghrelin and a therapeutic analogue called AZP-531 is explored.This is an interesting paper, on a very niche and underexplored hormonal pathway. The paper is well written and the figures are clear and convincing. Overall, the work is well conducted and the conclusions are valid, but a few issues need to be addressed.- I understand the rationale behind expressing B-Raf mutants in MCF7 cells to assess the altered response to unacylated ghrelin, but what are the basal physiological effects of this on the MCF7 model? Does it change estrogen dependence or basal cell viability/phenotype? Given that B-Raf mutations are not associated with ER+ disease, forcing a non-relevant mutant into a cell line model, likely has profound basal changes that will confound the experimental questions.

This is a very interesting point. As shown below, we have reanalyzed Figure 1D of the manuscript to examine the degree of stimulation by both E2 and FBS in cells transfected with mutant BRAF vs control vector by normalizing data to vehicle control treatment. We believe that this approach provides information related cell behavior, i.e. growth in response to stimuli, while retaining the key message related to response to UAG. As such, we have replaced Figure 1D in the manuscript with the revised Figure. Interestingly, while cells maintain similar response to growth factors (FBS), their response to estradiol is diminished. That said, cells do remain responsive to estradiol and UAG is unable to suppress either the estradiol- or growth factor-stimulated proliferation of mutant BRAF-transfected cells.

As a further morphological characterization of the cells, we examined the effect of mutant BRAF transfection on nuclear shape. No obvious differences were observed, e.g. elongation of nuclei seen with EMT, when compared to vector control-transfected cells.

**Author response image 1. sa2fig1:** 

- In the in vivo experiments, there was no mention of weight loss or potential physiological consequences of giving unacylated ghrelin (or analogues) on normal tissue functioning. Did the authors assess some of the standard 'health' indicators in this experiment?

Mice treated with unacylated ghrelin or AZP-531 did not lose weight compared to saline control (see new Supplementary raw data). We had two unacylated ghrelin-treated mice in the MCF7 cell xenograft study assessed by Australian Phenomics Network Histopathology and Organ Pathology, The University of Melbourne, and compared to vehicle control. No abnormalities specific to treatment were identified in Adrenal glands, Bladder, Brain, Cecum, Cervix, Clitoral gland, Colon, Duodenum, Eyes, Gall bladder, Harderian glands, Head, Heart, Hind leg (Long bone, Bone marrow, Synovial joint, Skeletal muscle), Ileum, Jejunum, Kidney, Liver, Lungs, Mesenteric lymph node, Ovaries, Oviducts, Pancreas, Salivary glands and Regional lymph nodes, Skin, Spinal cord, Spleen, Stomach, Tail, Thymus, Thyroids, Trachea, Uterus, and Vagina. These data are consistent with published data showing that unacylated ghrelin and AZP-531 are safe in rodents and humans. The following sentence was added in subsection “Unacylated ghrelin and cyclic analog AZP-531 inhibits tumor growth in xenograft models and patient-derived tumor cells” “Treatment with unacylated ghrelin had no detrimental effect on weight and no abnormalities associated with treatment were observed with histopathology assessment of a subset of mice.”

- The use of AZP-531 is justified because of improved chemical properties, but the assumption is that all the previous pathway work (on cAMP, MAPK etc) are valid for AZP-531. Is this the case or is there any evidence to support the conclusions that the mechanistic insight derived from unacylated ghrelin is directly applicable to the mode of action for AZP-531? it would be useful if even just one or two of the key functional endpoints explored with unacylated ghrelin, were validated with AZP-531.

This is a very important point raised by the reviewer. Ongoing studies in the laboratory in relation to characterization of AZP-531 mechanism of action demonstrate that similar to unacylated ghrelin (UAG), AZP-531 can suppress cAMP production from forskolin-stimulated MCF7 cells. We have included these new data (right) as Figure 6D in the manuscript. We have also added the following to subsection “Unacylated ghrelin analog, AZP-531, inhibits breast cancer cell growth in vitro, ex vivo and in vivo*”* “In order to confirm a similar mechanism of action, the effects of unacylated ghrelin and AZP-531 on cAMP levels within MCF7 cells were compared (Figure 6D). Both unacylated ghrelin and AZP-531 caused a dose-dependent inhibition of the forskolin-mediated production of cAMP at 100 and 1000 pM, with no significant differences observed when both treatments were compared.”

- The start of the Results section was confusing to a non-expert. The text states "The dependence on 3D culture was examined for unacylated ghrelin, where treatment with 100 pM resulted in suppression of cell growth in matrigel and collagen, but not in 2D (Figure 1—figure supplement 1A and 1B). At this dose, no effects on MDA-MB-231 cell growth were observed in either 2D or 3D cultures". It's not clear what cell model the first sentence refers to. I assumed it was the MDA-MB-231 cells, but the first line suggests there is growth inhibition in 3D, but the second sentence suggested there isn't growth inhibition in this cell line model.

We apologize for the lack of clarity. In subsection “Unacylated ghrelin inhibits the 3D growth of breast cancer cells”, we have replaced “The dependence on 3D culture was examined for unacylated ghrelin, where treatment with 100 pM resulted in suppression of cell growth in matrigel and collagen, but not in 2D (Figure 1—figure supplement 1A and 1B).” with “The dependence on 3D culture was examined for unacylated ghrelin, where treatment of MCF7 and MDA-MB-468 cells with 100 pM resulted in suppression of cell growth in matrigel and collagen, but not in 2D (Figure 1—figure supplement 1A and 1B).”